# GMCL1 controls 53BP1 stability and modulates taxane sensitivity

Yuki Kito[1,2], Tania J González-Robles[1,3,4], Sharon Kaisari[1,2,4], Juhee Pae[1,2,5†], Sheena Faye Garcia[1,2], Juliana Ortiz-Pacheco[1,2,6], Beatrix Ueberheide[1,2,6], Antonio Marzio[1,2*‡], Gergely Róna[1,2,4,7*], Michele Pagano[1,2,4*]

[1]Department of Biochemistry and Molecular Pharmacology, New York University Grossman School of Medicine, New York, United States; [2]Laura and Isaac Perlmutter Cancer Center, New York University Grossman School of Medicine, New York, United States; [3]Division of Precision Medicine, Department of Medicine, NYU Grossman School of Medicine, New York, United States; [4]Howard Hughes Medical Institute, NYU Grossman School of Medicine, New York, United States; [5]Department of Cell Biology, New York University Grossman School of Medicine, New York, United States; [6]Proteomics Laboratory, Division of Advanced Research Technologies, New York University Grossman School of Medicine, New York, United States; [7]MTA-HUN-REN RCNS Lendulet "Momentum" DNA Repair Research Group, Institute of Molecular Life Sciences, HUN-REN Research Centre for Natural Sciences, Budapest, Hungary

*For correspondence:
anm4031@med.cornell.edu (AM);
gergely.rona@nyulangone.org
(GR);
michele.pagano@nyulangone.
org (MP)

Present address: †Laboratory of Lymphocyte Dynamics, The Rockefeller University, New York, United States; ‡Department of Pathology and Laboratory Medicine, Meyer Cancer Center, Weill Cornell Medicine, New York, United States

## eLife Assessment

This study identifies 53BP1 as an interaction partner of GMCL1 (a likely CUL3 substrate receptor). The study proposes a novel mechanism by which cancer cells evade the mitotic surveillance pathway through GMCL1-mediated degradation of 53BP1, leading to reduced p53 activation and paclitaxel resistance. These data are the most **useful** aspect of the study, but the data supporting the authors' conclusions as to the clinical relevance of the study are **inadequate**. The authors have not taken relevant data about the clinical mechanism of taxanes into account.

**Abstract** Mitotic surveillance pathways monitor the duration of mitosis (M phase) in the cell cycle. Prolonged M phase, caused by spindle attachment defects or microtubule-targeting drugs, triggers formation of the ternary 'mitotic stopwatch pathway' complex (MSP) consisting of 53BP1, USP28, and p53. This complex stabilizes p53, leading to cell cycle arrest or apoptosis in daughter cells. In cancers that are resistant to paclitaxel, a microtubule-targeting agent, cells bypass mitotic surveillance activation, allowing unchecked proliferation, although the underlying mechanisms remain poorly understood. Here, we identify GMCL1 as a key negative regulator of MSP signaling. We show that 53BP1 physically interacts with GMCL1, but not its paralog GMCL2, and we map their interaction domains. CRL3[GMCL1] functions as a ubiquitin ligase that targets 53BP1 for degradation during the M phase, thereby reducing p53 accumulation in daughter cells. Depletion of GMCL1 inhibits cell cycle progression upon release from prolonged mitotic arrest, a defect that is rescued by co-silencing 53BP1 or USP28. Moreover, GMCL1 depletion sensitizes cancer cells to paclitaxel in a p53-dependent manner. Together, our findings support a model in which dysregulated CRL3[GMCL1]-mediated degradation of 53BP1 prevents proper MSP function, leading to p53 degradation and continued proliferation. Targeting GMCL1 may, therefore, represent one possible avenue for addressing paclitaxel resistance in cancer cells with functional p53.

## Introduction

Mitosis is orchestrated by several intracellular signaling pathways to ensure proper cell division while maintaining genomic integrity. Errors during cell division, including chromosome mis-segregation or spindle defects, could lead to changes in chromosome content, producing aneuploid, or polyploid progeny cells, which could be detrimental during development or lead to oncogenesis (*Hosea et al., 2024*; *Lens and Medema, 2019*; *Lambrus and Holland, 2017*). Therefore, cells have evolved quality control mechanisms to ensure proper cell division during M phase. One such surveillance mechanism is known as the Mitotic Stopwatch Pathway (MSP). Foundational work from the Sluder lab in 2010 (*Uetake and Sluder, 2010*) first demonstrated a p53-dependent G1 arrest following prolonged mitosis. This was later expanded by three key studies published in 2016, which identified USP28 (ubiquitin-specific protease 28) and 53BP1 (p53-binding protein 1) as critical components of the pathway (*Lambrus et al., 2016*; *Meitinger et al., 2016*; *Fong et al., 2016*). During prolonged mitosis without centrosome loss, a ternary complex of 53BP1, USP28, and p53 forms and persists into the G1 phase, where it induces p21 transcription and enforces p53-dependent cell cycle arrest. In this pathway, 53BP1 and USP28's deubiquitinase activity are required for p53 stabilization (*Stracker, 2024*; *Sparr and Meitinger, 2025*; *Meitinger et al., 2024*).

Proper mitotic arrest is critical for the efficacy of microtubule-targeting therapies, such as taxanes (e.g. paclitaxel and docetaxel), which disrupt spindle formation and chromosome segregation. Pharmacological disruption of mitosis can induce cell death (*Chan et al., 2012*), as observed with paclitaxel treatment (*Giannakakou et al., 2001*). Clinically, however, the effectiveness of taxanes is often limited, as many cancers develop resistance, including metastatic breast, ovarian, and non-small cell lung cancers (*Maloney et al., 2020*). This resistance is frequently associated with loss of MSP activity, for example, due to defective p53 signaling (*Gupta et al., 2019*; *Baird et al., 2010*; *Sosa Iglesias et al., 2018*). These observations underscore the urgent need to further elucidate the mechanisms underlying paclitaxel resistance in cancer.

Human germ cell-less protein-like 1 (GMCL1) is a putative substrate receptor of one of many CUL3-RING ubiquitin ligases (CRL3s). However, to date, the biological role of GMCL1 and its substrates have remained uncharacterized. *GMCL1* received its namesake from its *Drosophila melanogaster* homolog, *GCL* (Germ Cell-Less), which plays essential roles in early embryonic development and germ cell determination (*Jongens et al., 1994*; *Jongens et al., 1992*). The role of *GMCL1* in germ cell development appears to have been evolutionarily conserved as loss of *GMCL1* expression in mice has been shown to cause defects in meiosis and spermiogenesis (*Liebe et al., 2006*), and altered *GMCL1* expression was functionally associated with human asthenozoospermia (*Liu et al., 2018*). Although *GMCL1* homologs have been primarily associated with germ cell biology, several databases (e.g. GTEx *Aguet et al., 2020* and GENT2 *Park et al., 2019*) indicate that *GMCL1* is expressed in somatic cells as well, suggesting broader biological functions beyond germ cell development. In contrast, the GMCL1 paralog GMCL2 is specifically expressed in germ cells.

Here, we present studies that clarify the role of GMCL1 in somatic cells. We show that, similar to its *D. melanogaster* counterpart (*Pae et al., 2017*), GMCL1 functions as a CRL3 substrate receptor. Furthermore, we identify 53BP1 as a *bona fide* substrate of CRL3$^{GMCL1}$ and demonstrate that its levels are regulated by GMCL1 during prolonged mitotic arrest. By reducing mitotic 53BP1, GMCL1 inhibits the function of the USP28-p53-53BP1 mitotic stopwatch complex and limits p53 transmission to daughter cells. Based on these findings, we propose that GMCL1 inhibition may represent a potential novel approach to overcoming paclitaxel resistance in cancer cells with functional p53.

## Results

### Identification of 53BP1 as an interactor of GMCL1

We have shown that the GMCL1 ortholog in *D. melanogaster*, GCL, is a substrate recognition subunit of a CRL3 complex that is active specifically in mitosis (*Pae et al., 2017*). Therefore, we predicted the human GMCL1 to also behave as a CRL3 substrate receptor. GMCL1 contains a BTB (Broad-Complex, Tramtrack, and Bric-à-brac) and a BACK (BTB and C-terminal Kelch) domain (*Figure 1— figure supplement 1A*), consistent with other CRL3 receptors. On its C-terminus, GMCL1 also contains a GCL domain (residues 379–515), which is distinct from Kelch domains commonly found in other CUL3 substrate receptors. This GCL domain is predicted to form a β-sandwich characterized

by two opposing anti-parallel β-sheets, each made up of four β-strands (*Pae et al., 2017*; *Bonchuk et al., 2023*; *Figure 1—figure supplement 1A*). A Dali search (*Holm, 2022*) [1] of the GMCL1 C-terminal domain reveals that it has some structural homology to the MATH (meprin and TRAF homology) domain found in another CRL3 substrate receptor, SPOP (*Usher et al., 2021*), suggesting that the GCL domain in GMCL1 could potentially act as a protein-protein interaction motif to recruit substrates.

To investigate the role of GMCL1 in somatic cells, we used immunoprecipitation followed by mass spectrometry (IP-MS) to identify binding partners of GMCL1. Proteomics studies were performed by expressing and purifying the following FLAG-tagged proteins: (*i*) wild-type GMCL1 (GMCL1 WT), (*ii*) GMCL1 E142K (GMCL1 EK), which carries a mutation in the BTB domain that is predicted to disrupt its interaction with CUL3, and (*iii*) GMCL1 BTB/BACK-only (GMCL1 BBO) that lacks the GCL domain (*Figure 1A–C*). IP-MS analysis identified 1,765 potential binding partners that specifically interact with GMCL1 *via* its C-terminal domain. Using SAINT scores >0.70 and FDR <5%, this list was refined to 9 proteins that showed significant interaction with GMCL1 WT and GMCL1 EK, but not GMCL1 BBO, nominating 53BP1 as the most enriched protein (*Figure 1B*, *Supplementary file 1*). This selective enrichment suggests that the C-terminal, 'MATH-like' GCL domain of GMCL1 is critical for its interaction with binding partners, including 53BP1. The interaction between GMCL1 and endogenous 53BP1 and CUL3 was validated using immunoprecipitations (IPs) followed by immunoblotting (*Figure 1C and D*).

To further study the direct interaction between GMCL1 and 53BP1, we used AlphaFold 3 (*Abramson et al., 2024*) to locate the positions of contact residues at their predicted binding interface. Consistent with our initial IP-MS experiment, Alphafold 3 model predicted that the C-terminal domain of GMCL1 would interact with 53BP1. Based on the predicted GMCL1-53BP1 complex structure, we identified Arg 433, which appears to be a solvent-exposed residue in GMCL1's GCL domain that could interact with 53BP1 without impeding GMCL1 binding with CUL3. Thus, we generated the GMCL1 R433A (GMCL1 RA) point mutant and tested its binding to 53BP1 upon IP. As anticipated, compared to WT GMCL1, the R433A mutation completely abolished the binding of GMCL1 to 53BP1 but did not impact GMCL1's binding to CUL3 (*Figure 1D*).

To determine which region of 53BP1 mediates its binding to GMCL1, we mapped the predicted GMCL1-binding site on 53BP1 and identified a conserved IEDI amino acid sequence within the Minimal Focus Forming region (MFF) of 53BP1 (*Panier and Boulton, 2014*; *Figure 1—figure supplement 1B–E*). Through a series of IPs, we demonstrate that a 53BP1 mutant either lacking the MFF region or containing the IEDI-to-AAAA mutation within this region lost its interaction with CRL3^GMCL1, suggesting that this conserved IEDI sequence within the 53BP1 MFF region forms a critical degron recognized by GMCL1 (*Figure 1E*). To examine the interaction between endogenous GMCL1 and endogenous 53BP1, we used CRISPR-Cas9 to knock in a FLAG tag at the C-terminus of *GMCL1*. Immunoprecipitation experiments confirmed that endogenous GMCL1 interacts with both endogenous CUL3 and 53BP1 (*Figure 1F*).

Finally, we sought to determine whether GMCL2, a GMCL1 paralog, is also able to interact with 53BP1. Immunoprecipitation of FLAG-tagged GMCL1 or GMCL2 from HEK293T cells revealed that while GMCL1 binds to 53BP1, GMCL2 does not (*Figure 1—figure supplement 1F*), suggesting that GMCL1 and GMCL2 have distinct functions. Overall, our results suggest that GMCL1 is a CRL3 substrate receptor that interacts with 53BP1.

## 53BP1 is a *bona fide* substrate of GMCL1

To investigate whether GMCL1 regulates 53BP1 stability, we generated *GMCL1* knock-out (KO) cells using CRISPR Cas-9 (*Cong et al., 2013*) and compared 53BP1 levels between *GMCL1* WT and two *GMCL1* KO clones. RT-PCR analysis confirmed the loss of GMCL1 mRNA expression in the KO clones (*Figure 1—figure supplement 1G*). We found that 53BP1 levels were significantly increased in our *GMCL1* KO cells during M phase. While the whole cell extracts (WCE) showed modest differences in GMCL1 levels between the *GMCL1* WT and KO clones, our fractionation experiments revealed that both 53BP1 and p53 mainly accumulated in the chromatin-bound fraction of *GMCL1* KO M phase cells, and this increase did not correspond to a decrease in 53BP1 levels in the soluble fraction (*Figure 2A*).

To further explore the role of GMCL1 on 53BP1 stability, we stably reconstituted U2OS *GMCL1* KO cells with either GMCL1 WT or binding mutants (i.e. GMCL1 EK and GMCL1 RA). Notably, the

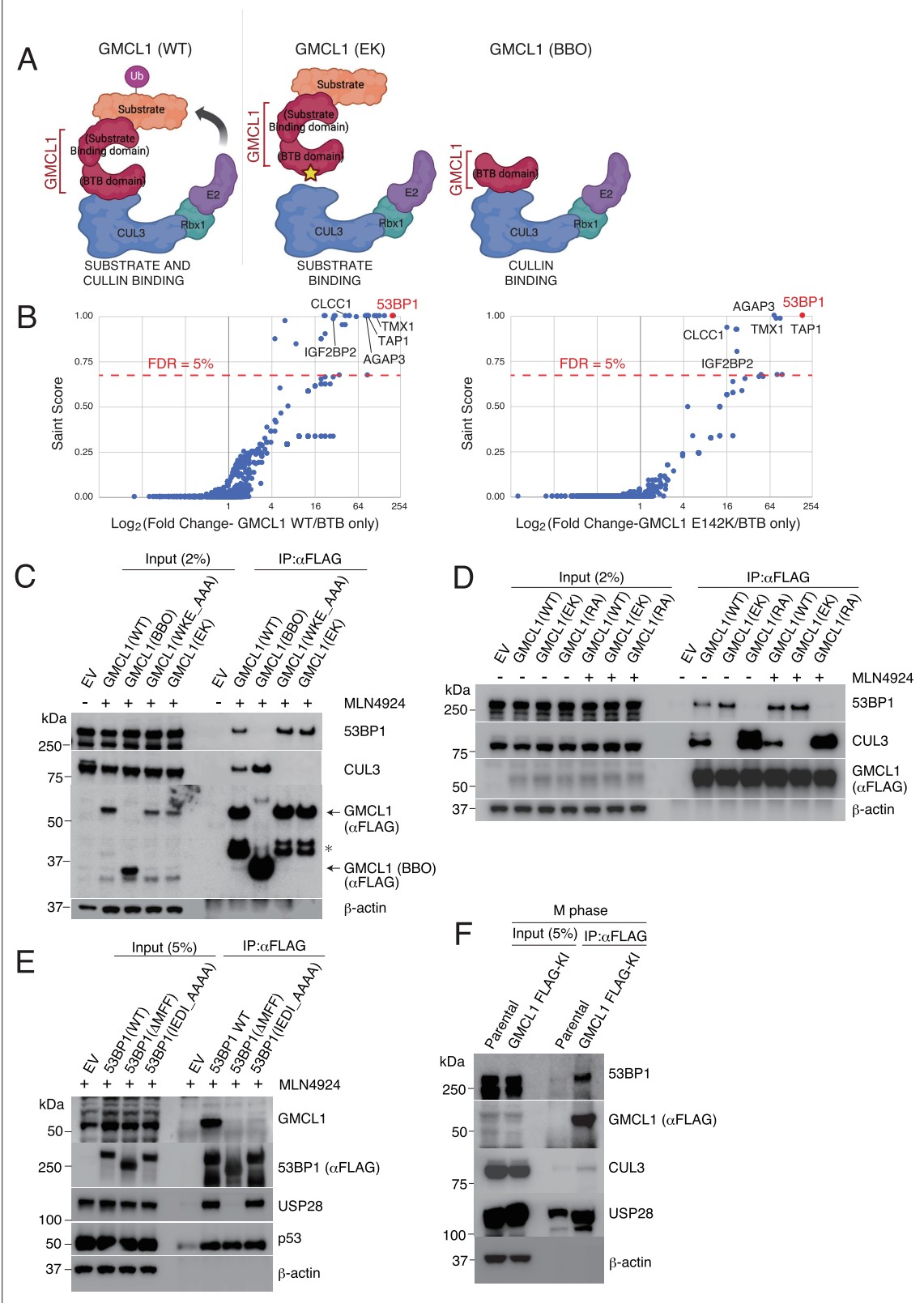

**Figure 1.** Identification of 53BP1 as a germ cell-less protein-like 1 (GMCL1) interactor. (**A**) Schematics for the immunoprecipitation-mass spectrometry (IP-MS) workflow using wild-type GMCL1 (GMCL1 WT) and mutants (GMCL1 EK and GMCL1 BBO). Color coding: Red, GMCL1; orange, putative substrates/interacting partners; blue, CUL3; green, RBX1; purple, E2 ubiquitin-conjugating enzyme. (**B**) HEK293T cells were transfected with FLAG-GMCL1 WT, FLAG-GMCL1 EK, or FLAG-GMCL1 BBO. After 24 hr, FLAG-tagged proteins were immunoprecipitated and analyzed by MS/MS. Left panel:

*Figure 1 continued on next page*

*Figure 1 continued*

proteins enriched with GMCL1 WT vs. BBO; right panel: proteins enriched with GMCL1 EK vs. BBO. Significant interactors were identified using SAINT scores >0.70 and FDR <5%. (**C**) HEK293T cells transfected with empty vector (EV), FLAG-GMCL1 WT, FLAG-GMCL1 BBO, FLAG-GMCL1 WKE_AAA (broadly disrupts the binding to CUL3), and FLAG-GMCL1 EK were treated with MLN4924 (3 hr). 53BP1 and CUL3 were immunoprecipitated with FLAG beads and analyzed by western blot. Asterisk indicates non-specific bands. This experiment was performed four times, and a representative blot is shown. (**D**) HEK293T cells were transfected with EV, FLAG-GMCL1 WT, FLAG-GMCL1 EK, or FLAG-GMCL1 RA. FLAG immunoprecipitations were probed for 53BP1 and CUL3. This experiment was performed four times, and a representative blot is shown. (**E**) HEK293T cells were transfected with EV, FLAG-53BP1 WT, FLAG-53BP1 ΔMFF, and FLAG-53BP1 IEDI_AAAA. After MLN4924 treatment (3 hr), 53BP1 was immunoprecipitated and immunoblotted. This experiment was performed three times, and a representative blot is shown. (**F**) M phase-synchronized GMCL1 FLAG knock-in HCT116 cells were collected. GMCL1 was immunoprecipitated using FLAG-beads and analyzed by immunoblotting.

The online version of this article includes the following source data and figure supplement(s) for figure 1:

**Source data 1.** This file contains the uncropped, unprocessed original blot images.

**Source data 2.** This file contains images with the regions trimmed for figure display marked in red.

**Figure supplement 1.** Mapping 53BP1 binding sites on germ cell-less protein-like 1 (GMCL1).

**Figure supplement 1—source data 1.** This file contains the uncropped, unprocessed original blot images.

**Figure supplement 1—source data 2.** This source data file contains full, uncropped western blot images, with red boxes marking the regions used in the corresponding figures.

accumulation of 53BP1 in *GMCL1* KO cells was rescued only upon re-expression of GMCL1 WT in the chromatin fraction. In contrast, both 53BP1 and p53 levels remained high in *GMCL1* KO cells expressing either GMCL1 EK or GMCL1 RA (*Figure 2B*, *Figure 2—figure supplement 1A*), emphasizing the importance of GMCL1's ability to bind both CUL3 (through the E142 residue) and 53BP1 (through the R433 residue) to regulate 53BP1 levels. Of note, the R433A mutant was expressed at levels comparable to the WT protein. Interestingly, the E142K mutant showed reduced expression in mitotic cells, yet was the most abundantly expressed in asynchronous cells. Decreases in 53BP1 protein observed upon GMCL1 WT expression was not accompanied by an increase in the soluble fraction (*Figure 2B*, *Figure 2—figure supplement 1A*), indicating that the reduction in chromatin-associated 53BP1 is not due to re-localization.

Next, we performed cycloheximide (CHX) to inhibit protein synthesis and to directly assess 53BP1 stability. In the chromatin-bound fraction, 53BP1 was more stable in *GMCL1* KO cells compared to cells rescued with GMCL1 WT (*Figure 2C*). To further support the direct role of GMCL1 in regulating 53BP1 turnover, we co-expressed FLAG-tagged Trypsin-Resistant Tandem Ubiquitin-Binding Entity (TR-TUBE), a construct composed of tandem ubiquitin-binding motifs (*Yoshida et al., 2015*), together with either GMCL1 WT or the E142K mutant in HEK293T cells. TR-TUBE efficiently pulled down endogenous ubiquitinated 53BP1, with a marked increase in the presence of GMCL1 WT, but not the GMCL1 E142K mutant (*Figure 2D* and *Figure 2—figure supplement 1B*). Collectively, these findings indicate that GMCL1 promotes 53BP1 ubiquitination and subsequent degradation during prolonged mitosis. Consequently, GMCL1-deficient cells retain elevated levels of 53BP1 and p53 following mitosis stress.

## GMCL1 regulation of MSP affects cell cycle progression in daughter cells

During mitosis, 53BP1 helps monitor centrosome integrity and mitotic progression through the MSP complex (53BP1-USP28-p53), transmitting this information into daughter cells (*Meitinger et al., 2024*). To investigate the effects of GMCL1 regulation on the MSP complex, we first analyzed post-mitotic *GMCL1* KO daughter cells 7 hr after release from prolonged nocodazole arrest. *GMCL1* KO daughter cells reconstituted with GMCL1 WT, exhibited low levels of 53BP1, p53, and p21, along with reduced expression of apoptosis-related genes (*Figure 2—figure supplement 1C, D*). In contrast, cells reconstituted with either the E142K or the R433A mutant displayed persistently elevated levels of 53BP1, p53, and p21, accompanied by increased expression of apoptosis-related genes (*Figure 2—figure supplement 1C, D*).

To assess how GMCL1 levels affect cell cycle progression following mitotic delays, we performed FACS analyses of EdU incorporation in hTERT-RPE1 cells. Cells subjected to extended mitotic arrest were released by mitotic shake-off into fresh medium. After 30 hr, significantly less cells were found

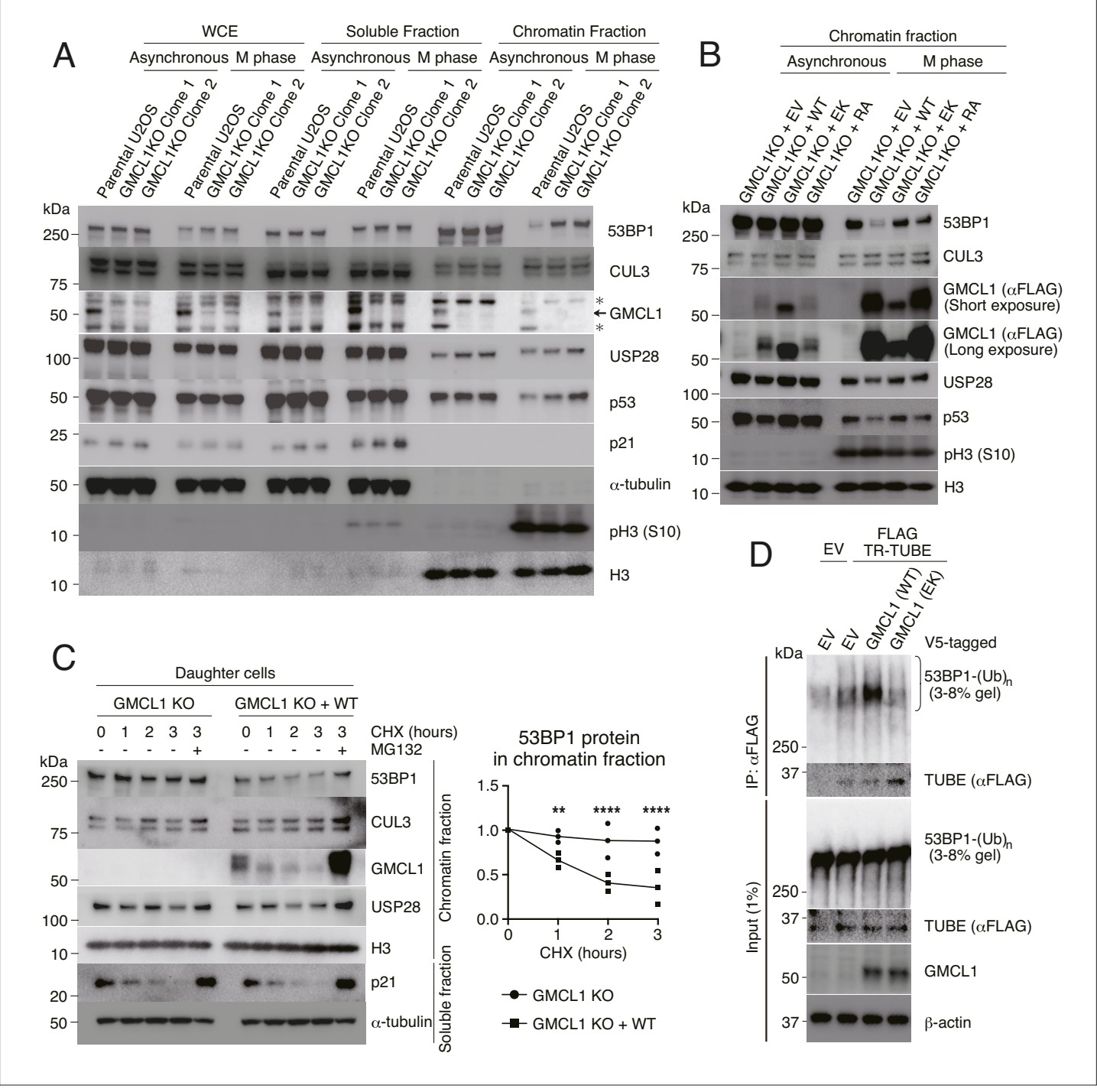

**Figure 2.** Germ cell-less protein-like 1 (GMCL1) targets 53BP1 for degradation during M phase. (**A**) Asynchronous or M phase-synchronized parental or *GMCL1* knockout (KO) U2OS cells were collected. Whole-cell extracts (WCE) were prepared using RIPA buffer, and other lysates were fractionated into soluble and chromatin-bound fractions for immunoblotting. Arrow indicates GMCL1-specific bands. Asterisk indicates non-specific bands. (**B**) Stable U2OS cell lines expressing empty vector (EV), FLAG-GMCL1 WT, FLAG-GMCL1 EK, or FLAG-GMCL1 RA in a *GMCL1* KO background were synchronized into M phase and fractionated. Immunoblots show the chromatin fraction. (**C**) M phase synchronized FLAG-GMCL1-expressing U2OS cells were collected by mitotic shake-off and cultured in fresh fetal bovine serum (FBS)-containing medium for 7 hr. G1 phase daughter cells were treated with cycloheximide (CHX) for the indicated time points, fractionated into soluble and chromatin-bound fractions, and analyzed by immunoblotting. Immunoblots show the chromatin fraction. Independent experiments were performed in triplicate. Differences between knockout (KO) and KO + WT were tested by two-way ANOVA followed by a Bonferroni test (**$p<0.005$, ****$p<0.0001$). (**D**) HEK293T cells were transfected with EV, V5-GMCL1 WT, or V5-GMCL1 EK, together with FLAG-TR-TUBE where indicated. FLAG immunoprecipitates were probed for 53BP1 and GMCL1. To visualize

*Figure 2 continued on next page*

Figure 2 continued

ubiquitinated 53BP1, the 53BP1 blot was resolved on a 3–8% gel. The arrow indicates the band corresponding to Trypsin-Resistant Tandem Ubiquitin-Binding Entity (TR-TUBE).

The online version of this article includes the following source data and figure supplement(s) for figure 2:

**Source data 1.** This file contains the uncropped, unprocessed original blot images.

**Source data 2.** This source data file contains full, uncropped western blot images, with red boxes marking the regions used in the corresponding figures.

**Figure supplement 1.** Mitotic stress imprints apoptotic memory in daughter cells.

**Figure supplement 1—source data 1.** This file contains the uncropped, unprocessed original blot images.

**Figure supplement 1—source data 2.** This source data file contains full, uncropped western blot images, with red boxes marking the regions used in the corresponding figures.

in S phase in the knockdown population, compared to control cells (*Figure 3A and B* and *Figure 3—figure supplement 1A*). Importantly, co-depletion of *GMCL1* with either *TP53BP1* (hereafter *53BP1*) or *USP28* abolished this G1 arrest, and cells proceeded into S phase at levels comparable to or even exceeding those of control cells, effectively rescuing the phenotype caused by *GMCL1* knockdown (*Figure 3A and B* and *Figure 3—figure supplement 1A*).

To further support our model, we overexpressed GMCL1 in hTERT-RPE1 cells and monitored cell cycle re-entry after mitotic delay. GMCL1-overexpressing cells progressed into S phase more rapidly than control cells, with notable entry observed as early as 18 hr post-release (*Figure 3C and D* and *Figure 3—figure supplement 1B*). Importantly, in the absence of anti-mitotic drug treatment, *GMCL1* KO cells exhibited no significant changes in baseline cell cycle profiles, regardless of reconstitution with WT or mutant GMCL1, compared to parental cells (*Figure 3—figure supplement 1C*).

Together, these findings identify GMCL1 as a key regulator of cell fate under mitotic stress by acting upstream of the USP28-p53-53BP1 axis to influence cell cycle re-entry dynamics.

## GMCL1 modulates taxane resistance in cancer cells

Taxanes, including paclitaxel, docetaxel, and cabazitaxel, are widely used chemotherapeutics that stabilize microtubules by preventing depolymerization (*Schiff et al., 1979*). However, resistance to taxanes commonly emerges through multiple mechanisms (*Maloney et al., 2020*; *Xu et al., 2023*; *Mosca et al., 2021*), including activation of pro-survival pathways, such as destabilization of the 53BP1-USP28-p53 complex. Given our finding that GMCL1 controls 53BP1 stability during prolonged mitosis, we sought to investigate whether GMCL1 expression is associated with taxane resistance and 53BP1 protein abundance in cancer cells. To this end, we leveraged the PRISM (profiling relative inhibition simultaneously in mixtures) repurposing dataset, which quantifies the proliferation-inhibitory effects of 4518 compounds across 578 cancer cell lines (*Corsello et al., 2020*), and integrated these data with DepMap proteomic and transcriptomic profiles (https://depmap.org). Since GMCL1 protein levels were not quantified in the dataset, we used *GMCL1* mRNA expression as a surrogate to assess its association with resistance to taxanes (*Figure 4A*). Interestingly, we found that five cancer types (i.e. endometrial, breast, kidney, pancreas, and upper aerodigestive tract cancers) with high levels of GMCL1 mRNA exhibited significant resistance to paclitaxel, cabazitaxel, and/or docetaxel (*Figure 4B*). In contrast, cell lines derived from 17 other cancer tissues with high GMCL1 mRNA expression did not show such significant correlation (*Figure 4—figure supplement 1*). Across multiple cancer types (*Figure 4—figure supplement 2*), we observed that in lung cancer cells with wild-type p53, high *GMCL1* expression combined with low 53BP1 levels was associated with significantly increased resistance to cabazitaxel and paclitaxel compared with cells showing low *GMCL1* expression and high 53BP1 levels (*Figure 4C*). In contrast, this relationship was absent in p53-mutant lung cancer cells, where *GMCL1* status did not correlate with taxane resistance (*Figure 4C*).

To verify the impact of GMCL1 levels on paclitaxel sensitivity, we performed cell viability and apoptosis assays using cells with wild-type or mutant p53. Paclitaxel treatment was chosen to mimic the conditions reported in DepMap. In p53 wild-type cells (MCF7 and U2OS), paclitaxel treatment led to a significant reduction in cell viability and an increase in apoptosis in *GMCL1*-depleted cells compared to cells transfected with non-targeting control siRNA (*Figure 5A–D*). However, *GMCL1* knockdown did not affect cell viability or apoptosis in paclitaxel-treated cells with inactivated p53 (HeLa and

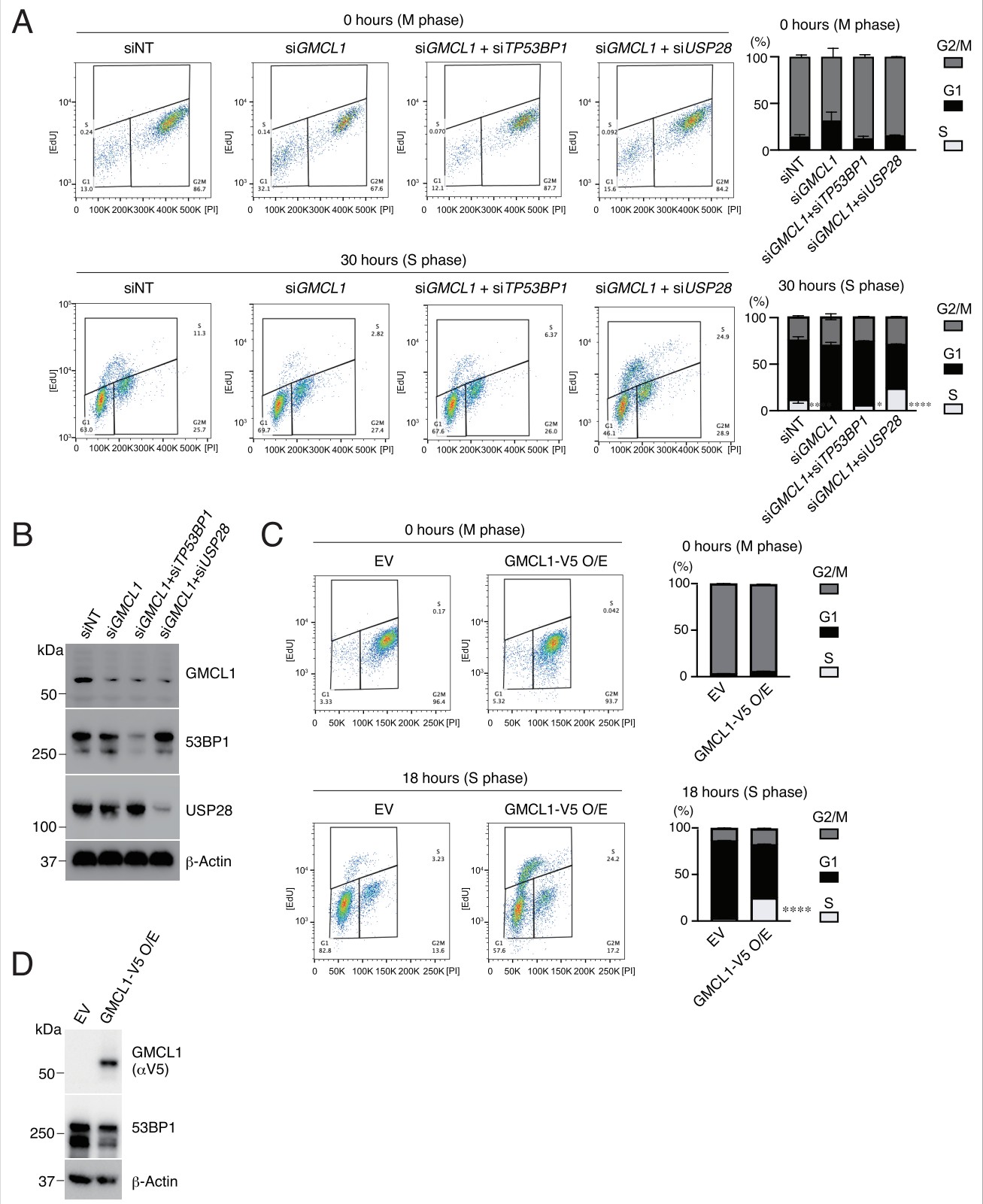

**Figure 3.** Cell cycle fate determination of daughter cells following prolonged mitosis. (**A**) Dot plots and graphs show the proportions of RPE1 cells in S, G1, and G2/M phases at the indicated time points following mitotic shake-off. Cells were synchronized in mitosis by nocodazole treatment for 16 hr and were subsequently released into fresh medium. Cell cycle distribution was determined by EdU pulse labeling and PI staining. EdU was added 1 hr prior to each indicated time point. Cells had been transfected 48 hr before the experiment with siNT, si*GMCL1*, or co-transfected with si*GMCL1* and either

*Figure 3 continued*

si*USP28* or si*TP53BP1*. Error bars represent standard deviation. Differences among four groups were tested by one-way ANOVA followed by Tukey's multiple comparisons test (*$p<0.05$, **$p<0.005$, ****$p<0.0001$). (**B**) Representative immunoblot showing the silencing efficiencies for panel (**A**). (**C**) Dot plots and graphs show the proportions of RPE1 cells stably expressing either an empty vector or V5-germ cell-less protein-like 1 (GMCL1), in S, G1, and G2/M phases at the indicated time points following mitotic shake-off. Cells were synchronized in mitosis by nocodazole treatment for 16 hr and were subsequently released into fresh medium. Cell cycle distribution was determined by EdU pulse labeling and PI staining. EdU was added 1 hr prior to each indicated time point. Error bars represent standard deviation. Differences between EV and V5-GMCL1 were tested by a two-tailed Welch's t test (****$p<0.0001$). (**D**) Representative immunoblot showing the overexpression of V5-GMCL1 for panel (**C**).

The online version of this article includes the following source data and figure supplement(s) for figure 3:

**Source data 1.** This file contains the uncropped, unprocessed original blot images.

**Source data 2.** This source data file contains full, uncropped western blot images, with red boxes marking the regions used in the corresponding figures.

**Figure supplement 1.** Cell cycle fate determination of daughter cells following prolonged mitosis in germ cell-less protein-like 1 (GMCL1) knockdown cells with 53BP1 or USP28.

HEC-1-A, respectively) (*Figure 5E–H*). Importantly, in hTERT-RPE1 cells, the reduction in cell viability and increase in apoptosis seen upon paclitaxel treatment in *GMCL1* knockdown cells were rescued by simultaneous knockdown of either *53* BP1 or *USP28* (*Figure 5I and J*). These observations are consistent with the results in *Figure 4* and suggest that paclitaxel resistance may, at least in part, be influenced by GMCL1 through the USP28-p53-53BP1 complex. Specifically, high GMCL1 expression appears to promote 53BP1 degradation, which in turn helps maintain lower p53 levels and reduces paclitaxel-induced cell death in cells with functional p53.

## Discussion

We identify GMCL1, a previously uncharacterized human CRL3 substrate receptor, as a regulatory component of the mitotic surveillance pathway. Specifically, we found that GMCL1 interacts with and mediates the degradation of 53BP1 during prolonged arrest in M phase. While 53BP1 is well known for its role in double-strand break (DSB) repair via non-homologous end joining (NHEJ) (*Zhao et al., 2020*), it also participates in the so-called mitotic stopwatch composed of the 53BP1-USP28-p53 complex that stabilizes p53 during prolonged mitotic arrest (*Lambrus et al., 2016*; *Meitinger et al., 2016*; *Fong et al., 2016*; *Meitinger et al., 2024*). We showed that GMCL1 controls the levels of 53BP1 and, consequently, those of p53 in mitotic cells, thereby influencing p53 transmission to daughter cells (see model in *Figure 5K*).

We found that GMCL1 primarily regulates the levels of chromatin-associated 53BP1. A PLK1-dependent 53BP1 re-localization from chromatin to the nucleoplasm has been reported in mitosis (*Meitinger et al., 2024*; *Burigotto et al., 2023*; *Lee et al., 2014*). However, multiple *GMCL1* KO clones display elevated chromatin-bound 53BP1 during M-phase arrest without a corresponding decrease in the soluble fraction, indicating the effect is not due to re-localization. The increased 53BP1 half-life in *GMCL1* KO daughter cells supports this conclusion. Discrepancies in the reported localization of 53BP1 (chromatin vs. nucleoplasm) during mitosis may reflect differences in biochemical fractionation methods (e.g. differences in the concentrations of salt or detergent and/or sonication conditions).

Our findings suggest that GMCL1 functions as a regulator of mitotic stress response, with potential oncogenic properties in certain contexts. This underscores the role of GMCL1 in mitotic regulation and chromosome stability, which may vary based on tumor type, genetic background, and additional oncogenic mutations. Accordingly, we observed that a subset of cell lines exhibiting resistance to taxane-based agents, such as paclitaxel, cabazitaxel, and docetaxel, displayed elevated *GMCL1* mRNA expression. The clinical application of our findings is currently limited by tumor heterogeneity and by the variable efficacy and cellular availability of taxanes during treatment.

GMCL1 has been primarily studied in the germ cells of *D. melanogaster*, where we have shown that it forms an active CRL3 complex during mitosis (*Pae et al., 2017*). Our new findings suggest a critical role for GMCL1 in mammalian somatic cell fate decisions by regulating 53BP1 stability during prolonged mitosis. We further show that GMCL1 loss sensitizes p53-wild-type, but not p53-mutant,

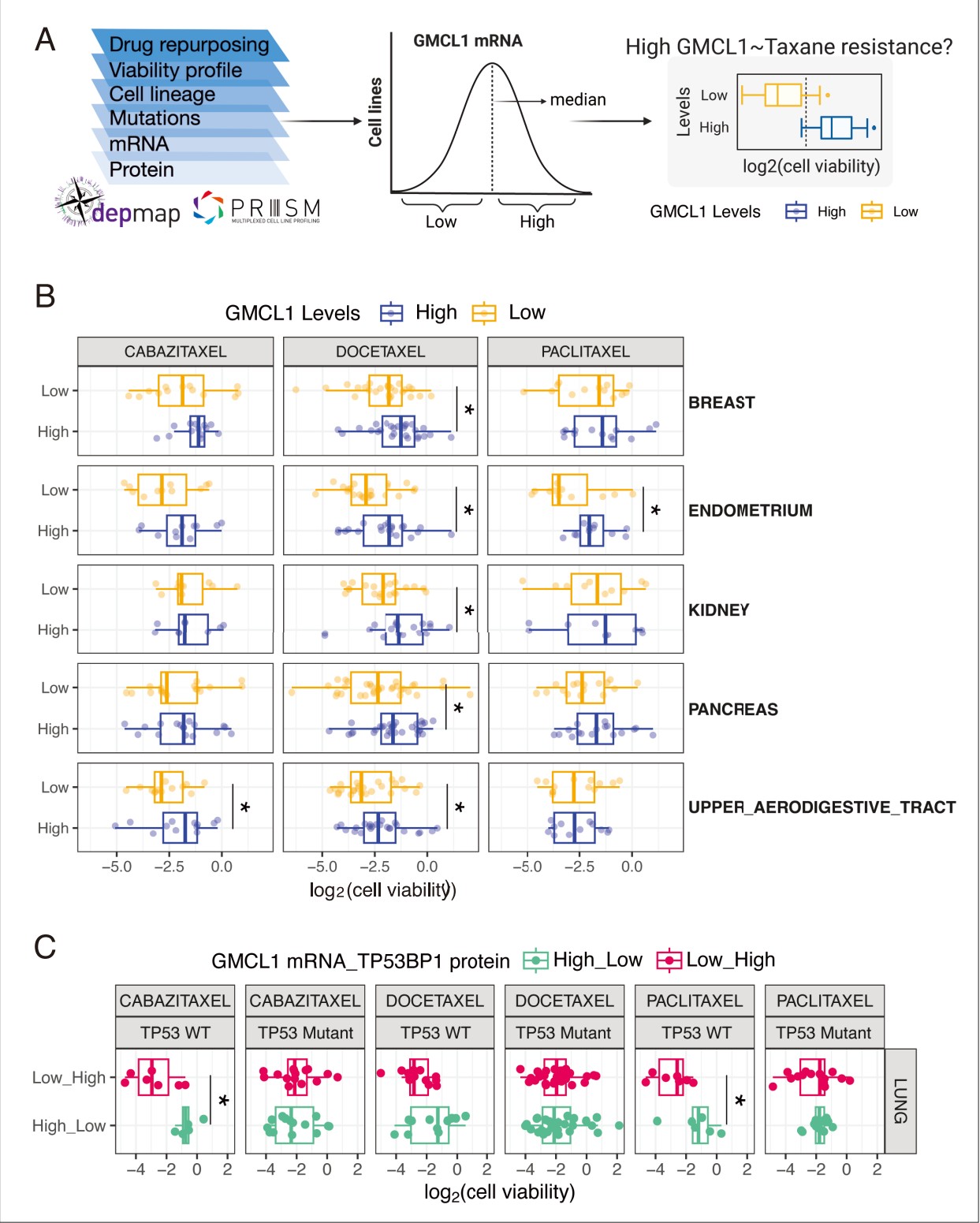

**Figure 4.** Germ cell-less protein-like 1 (GMCL1) expression shows positive correlation with taxane resistance in cancer cell lines. (**A**) Schematic overview of DepMap and PRISM data integration used in the analysis, including *GMCL1* mRNA (protein not available) and drug response for taxanes across DepMap cancer cell lines (left panel). Method for classification of cell lines into GMCL1-high and GMCL1-low groups based on its median mRNA expression levels within tissue types (middle panel). Schematic depicting comparison of taxane sensitivity between GMCL1-high and GMCL1-low groups (right panel). (**B**) Boxplots visualizing comparison of taxane sensitivity (i.e. cabazitaxel, docetaxel, and paclitaxel; log-fold change in cell viability) between GMCL1-high and GMCL1-low groups. Statistical significance was assessed using a two-sided, unpaired Wilcoxon rank sum test (*$p<0.05$).

*Figure 4 continued on next page*

*Figure 4 continued*

(**C**) Boxplots visualizing comparison of taxane sensitivity (i.e. cabazitaxel, docetaxel, and paclitaxel; log-fold change in cell viability) between *GMCL1* and *TP53BP1* High_Low and Low_High groups, respectively, further stratified by *TP53* mutation status. Statistical significance was assessed using a two-sided, unpaired Wilcoxon rank sum test (*$p<0.05$).

The online version of this article includes the following figure supplement(s) for figure 4:

**Figure supplement 1.** Association between GMCL1 expression levels and taxane sensitivity across cancer cell lines.

**Figure supplement 2.** TP53-status dependent correlations between GMCL1 and 53BP1 expression and taxane sensitivity.

cancer cells to paclitaxel-induced death, suggesting that GMCL1 inhibition may offer a selective therapeutic strategy for tumors with intact p53 function.

## Materials and methods

### Cell culture

Cell lines were purchased from ATCC and were routinely checked for mycoplasma contamination with the MycoStrip Mycoplasma Detection Kit (Invivogen). HEK293T (ATCC CRL-3216), HeLa (ATCC CCL-2), and hTERT RPE-1 (ATCC CRL-4000) were maintained in Dulbecco's modified Eagle's medium (DMEM) (Gibco). U-2 OS (ATCC HTB-96), HCT-116 (ATCC CCL-247), and HEC-1-A (ATCC HTB-112. NM) cells were maintained in McCoy's 5 A medium (Gibco). MCF7 (ATCC HTB-22) were maintained in Eagle's Minimum Essential Medium (EMEM) (ATCC). All media were supplemented with 10% fetal bovine serum (FBS) (Corning Life Sciences) and 1% penicillin/streptomycin/L-glutamine (Corning Life Sciences); however, MCF7 was further supplemented with human recombinant insulin (zinc solution; Gibco) to a final concentration of 11.2 µg/mL. All cell lines were maintained at 37 °C and 5% CO$_2$ in a humidified atmosphere.

### Plasmids, siRNA, and transfection

*Homo sapiens* cDNAs were amplified by PCR using KAPA HiFi DNA Polymerase (Kapa Biosystems) and sub-cloned into a variety of vector backbones, including modified pCDNA3.1 and pLVX-PURO lentiviral vectors containing C-terminal Flag, HA, or V5 tags. Site-directed mutagenesis was performed using KAPA HiFi DNA Polymerase (Kapa Biosystems). Plasmids were propagated in *E. coli* DH5α competent cells (New England Biolabs).

All cell lines were transiently transfected using Lipofectamine 3000 (Thermo Fisher Scientific) based on the manufacturer's recommendation. siRNA oligo transfections were performed using RNAiMax (Thermo Fisher Scientific) according to the manufacturer's instructions.

### Virus-mediated gene transfer

For the generation of lentivirus, HEK293T cells were transfected with pLVX constructs carrying the genes of interest, alongside the packaging plasmids pCMV-delta-R8.2 and pCMV-VSV-G. Viral supernatant was harvested 48 hr post-transfection, passed through a 0.45 µm sterile Millex-HV filter unit (Millipore Sigma), and supplemented with polybrene at a final concentration of 8 µg/ml (Sigma). Target cells were infected by replacing their culture medium with the virus-containing supernatant for an 8 hr incubation period. Selection of successfully transduced cells was performed using puromycin at a concentration of 1–2 µg/ml (Sigma).

### CRISPR-Cas9 genome editing

CRISPR-Cas9 genome editing techniques were carried out as previously described (*Rona et al., 2024*) with modifications. In brief, to generate *GMCL1*-knockout U-2 OS cells, optimal gRNA target sequences closest to the start codon of the genes were designed using the Benchling CRISPR Genome Engineering tool (https://www.benchling.com). For transient Cas9 expression, gRNAs specific to the *GMCL1* gene was incorporated into the pRP [CRISPR]-Hygro-hCas9-U6 vector, which was obtained from VectorBuilder (https://en.vectorbuilder.com/). The following oligos were used to generate the proper gRNA in the vector: GMCL1 (5'-CGTGCCCCCACGTACCTTCG-3'). To generate *GMCL1* 2x Flag knock-in HCT116 cells, an optimal gRNA target sequence closest to the genomic target site and a ~2 kb homologous recombination (HR) donor template was designed using the Benchling CRISPR

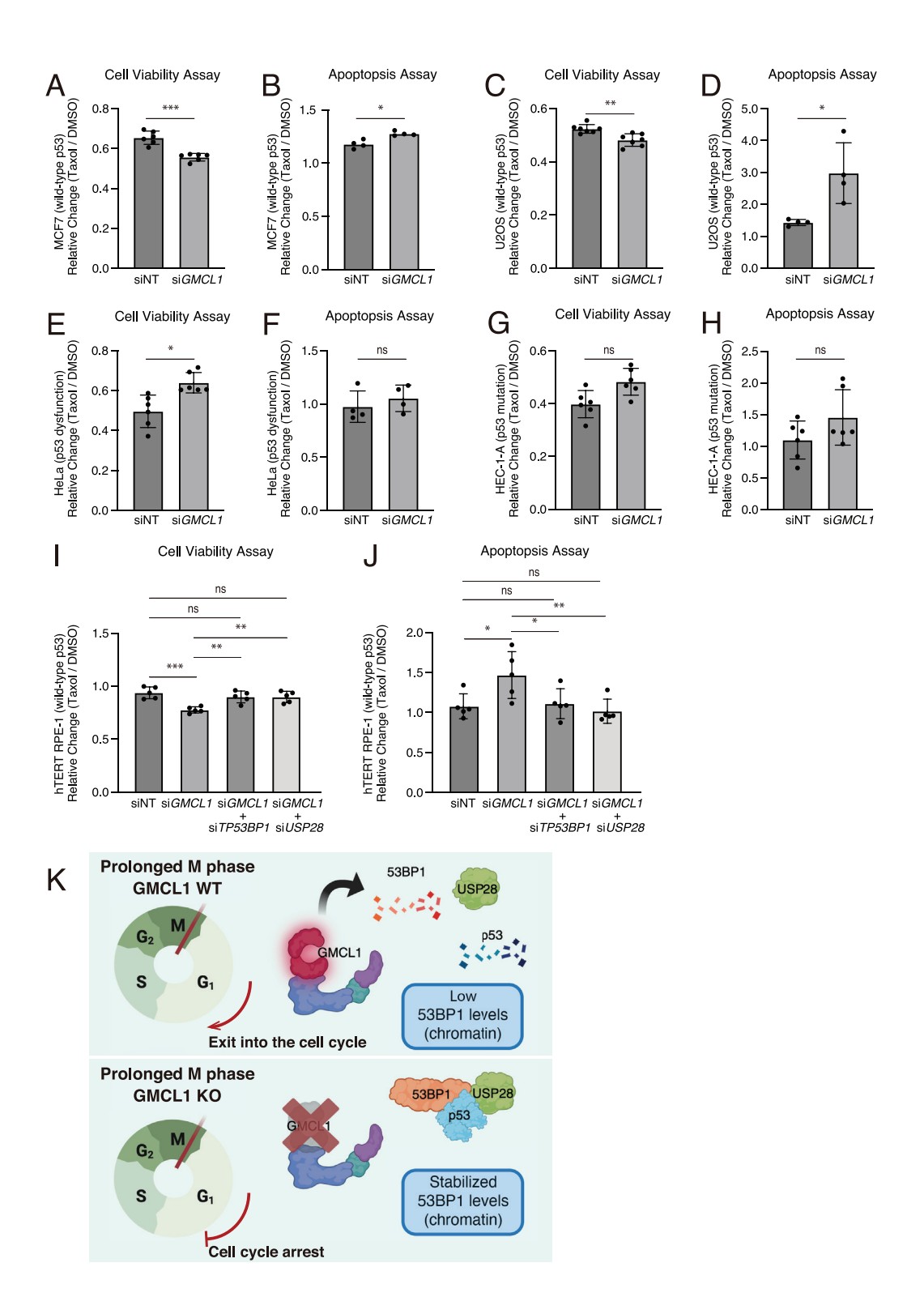

**Figure 5.** Germ cell-less protein-like 1 (GMCL1) deficiency sensitizes cancers with wild-type p53 to paclitaxel-induced apoptosis. (**A–G**) MCF7 (**A**), U2OS (**C**), HeLa (**E**), HEC-1-A (**G**) cells were transfected with GMCL1-targeting siRNAs or non-targeting (NT) control for 72 hr. Cells were treated with DMSO or 100 nM paclitaxel for 48 hr, and cell viability was assessed using the CellTiter-Glo Cell Viability Assay from four or six independent measurements. (**B–H**) Apoptosis was measured in the same conditions, i.e., MCF7 (**B**), U2OS (**D**), HeLa (**F**), HEC-1-A (**H**), using the RealTime-Glo Annexin V Apoptosis

*Figure 5 continued on next page*

*Figure 5 continued*

and Necrosis Assay from four or six independent measurements. For comparisons between two independent groups (**A–H**), a two-tailed Welch's test was applied (*$p<0.05$, **$p<0.005$, ***$p<0.001$). (**I**) hTERT-RPE1 cells were transfected with a non-targeting (NT) control or *GMCL1*-targeting siRNAs alone or in combination with either *TP53BP1* or *USP28* targeting siRNAs for 72 hr, followed by treatment with 100 nM paclitaxel for 48 hr. Cell viability was assessed using the CellTiter-Glo Cell Viability Assay from five independent measurements. (**J**) Apoptosis was measured in the same conditions, i.e., RPE1 using the RealTime-Glo Annexin V Apoptosis and Necrosis Assay from five independent measurements. Error bars represent standard deviation. For analysis involving four groups (**I and J**), one-way ANOVA followed by Tukey's multiple-comparisons test was applied (*$p<0.05$, **$p<0.005$, ***$p<0.001$). (**K**) Schematic model of this study. During prolonged mitosis, GMCL1 promotes degradation of 53BP1, thereby releasing p53 from the 53BP1-p53-USP28 ternary complex and leading to p53 degradation. As a result, daughter cells proceed through the cell cycle. In the absence of GMCL1, excessive accumulation of 53BP1 results in inheritance of the 53BP1-p53-USP28 ternary complex into daughter cells, where p21 expression is induced and cell cycle progression is arrested.

Genome Engineering tool. The HR donor template was designed to introduce a 2x Flag tag in frame with the C terminus of *GMCL1*, in the following order: GMCL1-linker-FLAG-linker-FLAG-Stop codon and was purchased from VectorBuilder (https://en.vectorbuilder.com/). The following single gRNA sequence was used for the transient hCas9 expression vector: GMCL1 (5'-AAGTTACAGCAGATAT ATAA-3').

Genomic DNA was collected using QuickExtract (Epicentre). Genotyping PCRs were performed with MyTaq HS Red Mix (Bioline), using primers surrounding the genomic target sites. The following primers were used for genotyping: GMCL1 (F: 5'-GCAGGCTTCTGATCTTCCCT-3', R: 5'-ACTTGTCA TCGTCGTCCTTGT-3'), and GMCL1 (F: 5'-GGGTGGGAGTTTGGAGAGTG-3', R:5'-TCTGGATTTTCT GGGTGACGA-3'). The resulting PCR products were then purified and sequenced to determine the presence of insertion or deletion events. Clones positive for insertion or deletion events were then validated by western blot.

## Antibodies

The following antibodies were used: β-actin (1:5000, Sigma-Aldrich A5441), CUL3 (1:1000, Bethyl Laboratories A301-109A), FLAG (1:2000, Sigma-Aldrich F7425), GMCL1 (1:1,000, Proteintech 15575–1-AP), HA (1:2000, Bethyl Laboratories A190-108A), Histone H3 (1:10,000, Abcam, ab1791), p21 (1:1000, Cell Signaling Technology 2947 S), p53 (1:1000, Proteintech 10442–1-AP), pHistone H3 (D2C8) (Ser10, 1:1000, Cell Signaling Technology, #3377), Ubiquitin (p37) (1:1000, Cell Signaling Technology, 58395 S), USP28 (1:1,000, Proteintech 17707–1-AP), 53BP1 (1:2000, Abcam ab36823), α-tubulin (1:5000, Sigma-Aldrich T6074).

## Drug treatment procedures

Where indicated, cells were treated with 400 ng/ml Nocodazole (Sigma-Aldrich M1404) for 16 hr, 100 nM paclitaxel for 48 hr, 10 μM MG132 for 3 hr, 2.5 μM MLN4924 for 3 hr, 100 μg/ml cycloheximide (CHX) for the indicated time.

## Cell synchronization

Cells were synchronized using nocodazole. Cells were treated with 100 ng/ml nocodazole for 14 hr. Mitotic cells were then collected by shake-off (M phase cells), or washed three times with PBS, and replated in normal medium to allow them to resume cell cycle (to analyze subsequent cell cycle progression in the daughter cells).

## qRT-PCR

Total RNA was purified using RNeasy mini kits (Qiagen). cDNA was generated using Double Primed EcoDry kits (Takara). The qPCR reaction was carried out using PowerUp SYBR Green (Applied Biosystems) and the Applied Biosystems QuantStudio 3 Real-Time PCR system in a 96-well format. ROX was used as a reference dye for fluorescent signal normalization and for well-to-well optical variations correction. Bar graphs represent the relative ratios of target genes to β-actin housekeeping gene values. For each biological sample, triplicate reactions were analyzed using absolute relative quantification method alongside in-experiment standard curves for each primer set to control for primer efficiency. The oligos used for qRT-PCR analysis were: β-actin (F: 5'- CATGTACGTTGCTATCCAGG C-3', R: 5'-CTCCTTAATGTCACGCACGAT-3'), GMCL1 (F: 5'-GGAGATTCCTGACCAGAACATTG-3',

R: 5′-CGACTGGGCTTTATCAAGACAT-3), GMCL2 (F: 5′-CCACGCAGCGGGTCTGT, R: 5′-TGGATTTT CTGGGTGACGATTATTT), p21 (F: 5′-TGTCCGTCAGAACCCATGC-3′, R: 5′-AAAGTCGAAGTTCCAT CGCTC-3′), NOXA (F: 5′-CCAAGCCGTGACCAAGGAC-3′, R: 5′-CGCCACATTGTGTAGCACCT-3′), PUMA delta (F: 5′- GCCAGATTTGTGAGACAAGAGG-3′, R: 5′- CAGGCACCTAATTGGGCTC-3′).

## Fractionation and immunoprecipitation

For whole-cell lysates, cells were directly lysed in a buffer containing 50 mM Tris-HCl (pH 7.5), 150 mM NaCl, 0.2% NP-40, 10% glycerol, 1 mM EDTA, 1 mM EGTA, 2 mM MgCl$_2$, and 1 mM dithiothreitol (DTT). Lysates were incubated on ice for 20 min and subsequently clarified by centrifugation at 20,000×g for 15 min at 4 °C. When cellular fractionation was performed, it followed a previously established method (*Rona et al., 2018*). Briefly, cells were lysed in CSK buffer (10 mM HEPES, pH 7.4, 100 mM NaCl, 300 mM sucrose, 0.1% Triton X-100, 3 mM MgCl$_2$, and 1 mM EGTA) for 5 min. The soluble fraction was collected by centrifugation at 1,300×g for 3 min at 4 °C. Cell pellets were subsequently washed in CSK buffer and then lysed in chromatin extraction buffer (50 mM Tris-HCl, pH 7.4, 250 mM NaCl, 0.1% Triton X-100, 1 mM EDTA, 50 mM NaF, 1 mM EGTA, 2 mM MgCl$_2$, and 250 U/mL Benzonase (Sigma-Aldrich)) for 30 min. Insoluble debris was removed by centrifugation at 20,000×g for 15 min at 4 °C. All buffers were supplemented with protease inhibitors (Complete ULTRA, Roche) and phosphatase inhibitors (Phosphatase Inhibitor Cocktail 2, Sigma-Aldrich).

For immunoprecipitation and affinity purification, samples were incubated with FLAG-M2 magnetic beads (Sigma-Aldrich) or anti-HA magnetic beads (Thermo Fisher Scientific) at 4 °C for 2 hr. Beads were thoroughly washed with wash buffer containing 50 mM Tris-HCl (pH 7.5), 150 mM NaCl, 0.2% NP-40, 1 mM EDTA, 1 mM EGTA, 2 mM MgCl$_2$, and 1 mM dithiothreitol (DTT), and protein elution was performed using either 3xFLAG peptide (Sigma-Aldrich) for mass spectrometry analysis or 1 x Laemmli sample buffer for Western Blot analysis.

## Immunoblotting

Western blotting was carried out as described previously (*Rona et al., 2024*). Protein samples were resolved under denaturing and reducing conditions on 4–12% Bis-Tris gels (NuPAGE) and transferred onto PVDF membranes (Immobilon-P, Millipore). Membranes were blocked with 5% nonfat dried milk, incubated overnight at 4 °C with primary antibodies, followed by washes and incubation with HRP-conjugated secondary antibodies (Amersham GE). Immunoreactive bands were visualized using enhanced chemiluminescence reagents (Pierce) and detected with a ChemiDoc MP imaging system (Bio-Rad). Each Western blot experiment was conducted at least three times to ensure reproducibility, with representative blots shown in the figures.

## Cell cycle analysis by flow cytometry

EdU incorporation and propidium iodide staining were performed either on asynchronous or synchronized cells. Visualization of EdU and propidium iodide staining was performed following the instructions of the manufacturer (*Rona et al., 2024*). In brief, cells were pulsed with EdU (10 μM), fixed and permeabilized, and EdU was detected by copper-free click chemistry using the Click-iT Plus EdU Alexa Fluor 488 Flow Cytometry Assay Kit (Thermo Fisher Scientific). Flow cytometry analysis of cell cycle distribution was conducted using a CytoFlex Analyzer (Beckman Coulter) and data were processed with FlowJo v10 software (Becton Dickinson).

## Mass spectrometry analysis of GMCL1 immunoprecipitations

The eluted anti-FLAG-tag antibody purified protein complexes were reduced with 2 μl of 0.2 M DTT for 1 hr at 57 °C and subsequently alkylated with 2 μl of 0.5 M iodoacetamide (Sigma) for 45 min at room temperature in the dark. 250 ng of SP3 beads (Cytiva) were added to proteins precipitated onto the beads by adding ethanol. Samples were placed in a thermomixer at 25 °C for 10 min. Beads were washed three times with 80% ethanol and then digested overnight with 400 ng of sequencing-grade modified trypsin (Promega) in 100 mM ammonium bicarbonate. Next, the samples were spun down at 21,000 × g for 1 min. The supernatant was transferred to a new tube while the beads were washed twice with 0.5% acetic acid. The washes were then combined with the supernatant collection. Samples were acidified with 10% TFA to pH 1 and loaded onto a 0.1% TFA equilibrated Pierce C18 spin column using a microcentrifuge. The samples were rinsed twice with 0.1% TFA and twice

more using 0.5% acetic acid. Peptides were eluted with 80% acetonitrile in 0.5% acetic acid. The organic solvent was removed using a SpeedVac concentrator, and the sample was reconstituted in 0.5% acetic acid.

An equal aliquot of each sample was loaded onto a trap column (Acclaim PepMap 100 pre-column, 75 µm×2 cm, C18, 3 µm, 100 Å, Thermo Scientific) connected to an analytical column (EASY-Spray column, 50 m×75 µm internal diameter, PepMap RSLC C18, 2 µm, 100 Å, Thermo Scientific) using the autosampler of an Easy nLC 1200 (Thermo Fisher Scientific) with solvent A consisting of 2% acetonitrile in 0.5% acetic acid and solvent B consisting of 80% acetonitrile in 0.5% acetic acid. The peptide mixture was gradient eluted using the following gradient: 5% solvent B for 5 min, 5–35% solvent B in 60 min, 35–45% solvent B in 10 min, followed by 45–100% solvent B in 10 min. The samples were acquired on the Orbitrap Eclipse using the following parameters: full MS spectra resolution of 120,000, an AGC target of 4e5, maximum ion time of 50 ms, scan range from 400 to 1500 m/z. The MS/MS spectra were collected with the following parameters: a resolution of 30,000, an AGC target of 2e5, maximum ion time of 30 ms, one microscan, 2 m/z isolation window, normalized collision energy (NCE) of 27, and a dynamic exclusion of 30 s. To identify binding partners, all acquired MS2 spectra were searched against a UniProt human database using Sequest HT within Proteome Discoverer 1.4 (Thermo Fisher Scientific). Fixed modifications were set on cysteine (carbamidomethyl), variable modifications of oxidation on methionine, and deamidation on glutamine and asparagine. The resulting peptide spectra match, and proteins are filtered to better than 1% false discovery rate (FDR), and only proteins with at least two different peptides are reported. Proteins differentially expressed between GMCL1 WT and EK, as determined by SAINT scores (*Choi et al., 2011*) with a 5% FDR, were considered significantly enriched interactions when comparing to GMCL1 BTB.

## Taxane resistance analysis

Taxane sensitivity data were obtained from the PRISM Repurposing dataset (DepMap 24Q2, https://depmap.org/), which reports log2-fold changes (LFC) in cell viability across 578 cancer cell lines treated with various compounds, including paclitaxel, cabazitaxel, and docetaxel (*Corsello et al., 2020*). Data preprocessing and integration: We integrated the GMCL1 RSEM-normalized mRNA expression (DepMap filename: OmicsExpressionProteinCodingGenesTPMLogp1BatchCorrected.csv), with TP53BP1 protein abundance (*Gonçalves et al., 2022*) and TP53 mutation status (DepMap filename: OmicsSomaticMutationsMatrixHotspot.csv) across all cell lines catalogued in the PRISM dataset into a harmonized dataset using R version 4.2.1 (*Supplementary file 2*). Cell lines with missing GMCL1 or TP53BP1 expression, taxane LFC data, or tissue type annotation were excluded from the analysis.

Stratification by expression levels: For tissue-level comparisons (e.g. breast, endometrium, kidney, pancreas, etc.), cell lines were stratified in parallel based on the median within tissue type of GMCL1 mRNA expression or TP53BP1 protein abundance into 'high' and 'low' GMCL1 or TP53BP1 groups, respectively. This allowed us to assess the relationship between baseline GMCL1 or TP53BP1 levels, TP53 mutation status, and taxane resistance under standardized, non-physiological screening conditions.

Statistical analysis: Differences in taxane sensitivity (LFC values) between high and low expression groups were assessed using two-sided, unpaired Wilcoxon rank-sum tests between 'high' vs 'low' groups (*$p<0.05$). To account for multiple testing across different tissue types and drug combinations, we applied the Benjamini-Hochberg false discovery rate (FDR). Importantly, the dataset does not specifically isolate M-phase cells, but rather represents mixed populations, and the findings should be interpreted accordingly.

## Cell viability and apoptosis assays

Cells (2500/well) were plated in a 96-well plate. The medium was replaced with 50 µL of medium containing the target siRNA (at a final concentration of 20 nM). After 24 hr, an additional 50 µL of medium containing either DMSO or paclitaxel (at a final concentration of 100 nM) was added. Luminescence was measured using BioTek Synergy Neo2 (Agilent) 48 hr post-treatment, using the Cell-Titer-Glo 2.0 Cell Viability Assay Kit (Promega) or RealTime-Glo Annexin V Apoptosis and Necrosis Assay Kit (Promega) following the manufacturer's recommendations.

## Quantification and statistical analysis

Data analysis was performed using GraphPad Prism version 10.2.1. For comparisons involving three or more groups, one-way ANOVA followed by Bonferroni's post hoc test or the Brown-Forsythe and Welch ANOVA test followed by Dunnett's T3 multiple comparisons test was applied.

## Acknowledgements

We would like to thank the NYU Proteomics Lab (supported in part by NYU School of Medicine and the Laura and Isaac Perlmutter Cancer Center Support grant P30CA016087 from the National Cancer Institute). We also thank the members of the Pagano Lab for helpful discussions. MP is an investigator with the Howard Hughes Medical Institute, and his laboratory is supported by grant R35-GM136250 from the NIH. YK is a recipient of the Japan Society for the Promotion of Science (JSPS) Postdoctoral Fellowships and the Uehara Memorial Foundation Postdoctoral fellowship. TGR is grateful for funding from NIH Institutional training grant in Cell Biology (T32GM136542) and HHMI Gilliam Fellowship (GT15758). SK is supported by the K99 Career Development Award from NIGMS (1K99GM155613-01A1) and has been a Life Sciences Research Foundation (LSRF) awardee and an EMBO Long Term Postdoctoral Fellow. AM is supported by the US Department of Defense (DoD) (HT9425-24-1-0019) and the National Cancer Institute (R01 CA296867-01A1). GR is supported by the Momentum Grant of the Hungarian Academy of Sciences (LP2023-15/2023), EMBO Installation Grant (IG5670-2024), and the HUN-REN Welcome Home and Foreign Researcher Recruitment Grant (KSZF-143/2023). Dr. Ruth Lehmann was included as a co-author in the original submission due to her intent to contribute to the manuscript's preparation, but her name was later removed when she was unable to participate.

## Additional information

### Competing interests

Michele Pagano: Is or has been an advisor for SEED Therapeutics, CullGen, Deargen, Kymera Therapeutics, Lumanity, Serinus Biosciences, Sibylla Biotech, Triana Biomedicines, and Umbra Therapeutics; also has financial interests in CullGen, Kymera Therapeutics, SEED Therapeutics, Thermo Fisher Scientific, and Triana Biomedicines. The other authors declare that no competing interests exist.

### Funding

| Funder | Grant reference number | Author |
|---|---|---|
| National Institutes of Health | R35-GM136250 | Michele Pagano |
| Japan Society for the Promotion of Science | | Yuki Kito |
| Uehara Memorial Foundation | | Yuki Kito |
| NIH Office of the Director | T32GM136542 | Tania J González-Robles |
| Howard Hughes Medical Institute | GT15758 | Tania J González-Robles |
| National Institute of General Medical Sciences | 1K99GM155613-01A1 | Sharon Kaisari |
| Life Sciences Research Foundation | | Sharon Kaisari |
| European Molecular Biology Organization | | Sharon Kaisari |
| United States Department of Defense | HT9425-24-1-0019 | Antonio Marzio |
| National Cancer Institute | R01 CA296867-01A1 | Antonio Marzio |

| Funder | Grant reference number | Author |
|---|---|---|
| Hungarian Academy of Sciences | LP2023-15/2023 | Gergely Róna |
| European Molecular Biology Organization | IG5670-2024 | Gergely Róna |
| Hungarian Research Network | KSZF-143/2023 | Gergely Róna |
| National Cancer Institute | P30CA016087 | Beatrix Ueberheide |

The funders had no role in study design, data collection and interpretation, or the decision to submit the work for publication.

## Author contributions

Yuki Kito, Conceptualization, Data curation, Formal analysis, Validation, Investigation, Visualization, Methodology, Writing – original draft, Writing – review and editing; Tania J González-Robles, Resources, Data curation, Software, Formal analysis, Validation, Investigation, Visualization, Methodology, Writing – original draft, Writing – review and editing; Sharon Kaisari, Validation, Investigation, Methodology, Writing – original draft, Writing – review and editing; Juhee Pae, Investigation, Methodology; Sheena Faye Garcia, Resources, Investigation, Visualization, Methodology, Writing – original draft; Juliana Ortiz-Pacheco, Beatrix Ueberheide, Resources, Investigation, Methodology; Antonio Marzio, Conceptualization, Supervision, Investigation, Visualization, Methodology, Project administration, Writing – review and editing; Gergely Róna, Conceptualization, Data curation, Formal analysis, Supervision, Validation, Investigation, Visualization, Methodology, Writing – original draft, Project administration, Writing – review and editing; Michele Pagano, Conceptualization, Data curation, Formal analysis, Supervision, Funding acquisition, Validation, Investigation, Visualization, Methodology, Writing – original draft, Project administration, Writing – review and editing

## Author ORCIDs

Yuki Kito ⓘ https://orcid.org/0000-0001-5104-0932
Tania J González-Robles ⓘ https://orcid.org/0000-0001-9292-382X
Sharon Kaisari ⓘ https://orcid.org/0000-0002-6884-3886
Beatrix Ueberheide ⓘ https://orcid.org/0000-0003-2512-0204
Antonio Marzio ⓘ https://orcid.org/0000-0001-5959-012X
Gergely Róna ⓘ https://orcid.org/0000-0003-3222-7261
Michele Pagano ⓘ https://orcid.org/0000-0003-3210-2442

Reviewer #2 (Public review): https://doi.org/10.7554/eLife.106730.3.sa1
Reviewer #3 (Public review): https://doi.org/10.7554/eLife.106730.3.sa2
Author response https://doi.org/10.7554/eLife.106730.3.sa3

# Additional files

## Supplementary files

Supplementary file 1. The table of proteins identified by IP-MS analysis with SAINT scores > 0.70 and FDR < 5%.

Supplementary file 2. Table integrating PRISM Repurposing drug sensitivity data with DepMap-derived GMCL1 RSEM-normalized mRNA expression, TP53BP1 protein abundance, and TP53 mutation status.

MDAR checklist

## Data availability

Original western blot images have been deposited at Mendeley at DOI:10.17632/gj3x6r263d.1 and are publicly available as of the date of publication. https://data.mendeley.com/preview/gj3x6r263d?a=1e452dfc-bf85-472c-83ce-0e997ba6fa40. The mass spectrometric raw files are accessible at https://massive.ucsd.edu under accession MassIVE MSV000097235 and at https://www.proteomexchange.org/ under accession PXD061458.

The following datasets were generated:

| Author(s) | Year | Dataset title | Dataset URL | Database and Identifier |
|---|---|---|---|---|
| Kito Y, González-Robles TJ, Kaisari S, Pae J, Garcia SF, Ortiz-Pacheco J, Ueberheide B, Marzio A, Róna G, Pagano M | 2025 | GMCL1 Controls 53BP1 Stability and Modulates Taxane Sensitivity | https://massive.ucsd.edu/ProteoSAFe/private-dataset.jsp?task=41d56cfeec4145fab90aa98bf4a6f33e | MassIVE MSV000097235, MSV000097235 |
| Kito Y, González-Robles TJ, Kaisari S, Pae J, Garcia SF, Ortiz-Pacheco J, Ueberheide B, Marzio A, Róna G, Pagano M | 2025 | GMCL1 Controls 53BP1 Stability and Modulates Taxane Sensitivity | https://doi.org/10.17632/gj3x6r263d.1 | Mendeley Data, 10.17632/gj3x6r263d.1 |
| Kito Y, González-Robles TJ, Kaisari S, Pae J, Garcia SF, Ortiz-Pacheco J, Ueberheide B, Marzio A, Róna G, Pagano M | 2025 | GMCL1 Controls 53BP1 Stability and Modulates Taxane Sensitivity | https://proteomecentral.proteomexchange.org/cgi/GetDataset?ID=PXD061458 | ProteomeXchange, PXD061458 |

The following previously published dataset was used:

| Author(s) | Year | Dataset title | Dataset URL | Database and Identifier |
|---|---|---|---|---|
| Gonçalves E, Poulos RC, Cai Z, Barthorpe S, Manda SS, Lucas N, Beck A, Bucio-Noble D, Dausmann M, Hall C | 2022 | Pan-cancer proteomic map of 949 human cell lines | https://www.ebi.ac.uk/pride/archive/projects/PXD030304 | PRIDE, PXD030304 |

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
