## [Editor Report · eLife Assessment]

This study identifies 53BP1 as an interaction partner of GMCL1 (a likely CUL3 substrate receptor). The study proposes a novel mechanism by which cancer cells evade the mitotic surveillance pathway through GMCL1-mediated degradation of 53BP1, leading to reduced p53 activation and paclitaxel resistance. These data are the most **useful** aspect of the study, but the data supporting the authors' conclusions as to the clinical relevance of the study are **inadequate**. The authors have not taken relevant data about the clinical mechanism of taxanes into account.

---

## [Referee Report · Reviewer #2 (Public review)]

Summary

This study investigates the role of GMCL1 in regulating the mitotic surveillance pathway (MSP), a protective mechanism that activates p53 following prolonged mitosis. The authors identify a physical interaction between 53BP1 and GMCL1, but not with GMCL2. They propose that the ubiquitin ligase complex CRL3-GMCL1 targets 53BP1 for degradation during mitosis, thereby preventing the formation of the "mitotic stopwatch" complex (53BP1-USP28-p53) and subsequent p53 activation. The authors show that high GMCL1 expression correlates with resistance to paclitaxel in cancer cell lines that express wild-type p53. Importantly, loss of GMCL1 restores paclitaxel sensitivity in these cells, but not in p53-deficient lines. They propose that GMCL1 overexpression enables cancer cells to bypass MSP-mediated p53 activation, promoting survival despite mitotic stress. Targeting GMCL1 may thus represent a therapeutic strategy to re-sensitize resistant tumors to taxane-based chemotherapy.

Strengths

This manuscript presents potentially interesting observations. The major strength of this article is the identification of GMCL1 as 53BP1 interaction partner. The authors identified relevant domains and show that GMCL1 controls 53BP1 stability. The authors further show a potentially interesting link between GMCL1 status and sensitivity to Taxol.

Weaknesses

A major limitation of the original manuscript was that the functional relevance of GMCL1 in regulating 53BP1 within an appropriate model system was not clearly demonstrated. In the revised version, the authors attempt to address this point. However, the new experiment is insufficiently controlled, making it difficult to interpret the results. State-of-the-art approaches would typically rely on single-cell tracking to monitor cell fate following release from a moderately prolonged mitosis.

In contrast, the authors use a population-based assay, but the reported rescue from arrest is minimal. If the assay were functioning robustly, one would expect that nearly all cells depleted of USP28 or 53BP1 should have entered S-phase at a defined time after release. Thus, the very small rescue effect of siTP53BP1 suggests that the current assay is not suitable. It is also likely that release from a 16-hour mitotic arrest induces defects independent of the 53BP1-dependent p53 response.

Furthermore, the cell-cycle duration of RPE1 cells is less than 20 hours. It is therefore unclear why cells are released for 30 hours before analysis. At this time point, many cells are likely to have progressed into the next cell cycle, making it impossible to draw conclusions regarding the immediate consequences of prolonged mitosis. As a result, the experiment cannot be evaluated due to inadequate controls.

To strengthen this part of the study, I recommend that the authors first establish an assay that reliably rescues the mitotic-arrest-induced G1 block upon depletion of p53, 53BP1, or USP28. Once this baseline is validated, GMCL1 knockout can then be introduced to quantify its contribution to the response.

A broader conceptual issue is that the evidence presented does not form a continuous line of reasoning. For example, it is not demonstrated that GMCL1 interacts with or regulates 53BP1 in RPE1 cells-the system in which the limited functional experiments are conducted.

There are also a number of inconsistencies and issues with data presentation that need to be addressed:

(1) Figure 2C: p21 levels appear identical between GMCL1 KO and WT rescue. If GMCL1 regulates p53 through 53BP1, p21 should be upregulated in the KO.

(2) Figure 2A vs. 2C: GMCL1 KO affects chromatin-bound 53BP1 in Figure 2A, yet in Figure 2C it affects 53BP1 levels specifically in G1-phase cells. This discrepancy requires clarification.

(3) Figure 2C quantification: The three biological repeats show an unusual pattern, with one repeat's data points lying exactly between the other two. It is unclear what the line represents; please clarify.

(4) Figure nomenclature: Some abbreviations (e.g., FLAG-KI in Fig. 1F, WKE in Fig. 1C-D, ΔMFF in Fig. 1E) are not defined in the figure legends. All abbreviations must be explained.

(5) Figure 2D: Please indicate how many times the experiment was reproduced. Quantification with statistical testing would strengthen the result. Pull-downs of 53BP1 with calculation of the ubiquitinated/total ratio could also support the conclusion.

(6) Figures 3A and 3C: The G1 bars share the same color as the error bars, making the graphs difficult to interpret. Please adjust the color scheme.

---

## [Referee Report · Reviewer #3 (Public review)]

Summary:

In this study, Kito et al follow up on previous work that identified Drosophila GCL as a mitotic substrate recognition subunit of a CUL3-RING ubiquitin ligase (CRL3) complex. Here they identified mutants of the human ortholog of GCL, GMCL1, that disrupt the interaction with CUL3 (GMCL1E142K) and that lack the substrate interaction domain (GMCL1 BBO). Immunoprecipitation followed by mass spectrometry identified 9 proteins that interacted with wild type FLAG-GMCL1 but not GMCL1 EK or GMCL1 BBO. These proteins included 53BP1, which plays a well characterized role in double strand break repair but also functions in a USP28-p53-53BP1 "mitotic stopwatch" complex that arrests the cell cycle after a substantially prolonged mitosis. Consistent with the IP-MS results, FLAG-GMCL1 immunoprecipitated 53BP1. Depletion of GMCL1 during mitotic arrest increased protein levels of 53BP1, and this could be rescued by wild type GMCL1 but not the E142K mutant or a R433A mutant that failed to immunoprecipitate 53BP1. Using a publicly available dataset, the authors identified a relatively small subset of cell lines with high levels of GMCL1 mRNA that were resistant to the taxanes paclitaxel, cabazitaxel, and/or docetaxel. This type of analysis is confounded by the fact that paclitaxel and other microtubule poisons accumulate to substantially different levels in various cell lines (PMID: 8105478, PMID: 10198049) so careful follow up experiments are required to validate results. The correlation between increased GMCL1 mRNA and taxane resistance was not observed in lung cancer cell lines. The authors propose this was because nearly half of lung cancers harbor p53 mutations, and lung cancer cell lines with wild type but not mutant p53 showed the correlation between increased GMCL1 mRNA and taxane resistance. However, the other cancer cell types in which they report increased GMCL1 expression correlates with taxane sensitivity also have high rates of p53 mutation. Furthermore, p53 status does not predict taxane response in patients (PMID: 10951339, PMID: 8826941, PMID: 10955790). The authors then depleted GMCL1 and reported that it increased apoptosis in two cell lines with wild type p53 (MCF7 and U2OS) due to activation of the mitotic stopwatch. This is surprising because the mitotic stopwatch paper cited (PMID: 38547292) reported that U2OS cells have an inactive stopwatch. Though it can be partially restored by treatment with an inhibitor of WIP1, the stopwatch was reported to be substantially impaired in U2OS cells, in contrast to what is reported here. Additionally, activation of the stopwatch results in cell cycle arrest rather than apoptosis in most cell types, including MCF7. Beyond this, it has recently been shown that the level of taxanes and other microtubule poisons achieved in patient tumors is too low to induce mitotic arrest (PMID: 24670687, PMID: 34516829, PMID: 37883329). Physiologically relevant concentrations are achieved with approximately 5-10 nM paclitaxel, rather than the 100 nM used here. The findings here demonstrating that GMCL1 mediates chromatin localization of 53BP1 during mitotic arrest are solid and of interest to cell biologists, but it is unlikely that these findings are relevant to paclitaxel response in patients.

Strengths:

This study identified 53BP1 as a target of CRL3GMCL1-mediated degradation during mitotic arrest. AlphaFold3 predictions of the binding interface followed by mutational analysis identified mutants of each protein (GMCL1 R433A and 53BP1 IEDI1422-1425AAAA) that disrupted their interaction. Knock-in of a FLAG tag into the C-terminus of GMCL1 in HCT116 cells followed by FLAG immunoprecipitation confirmed that endogenous GMCL1 interacts with endogenous CUL3 and 53BP1 during mitotic arrest.

Weaknesses:

The clinical relevance of the study is overinterpreted. The authors have not taken relevant data about the clinical mechanism of taxanes into account. Supraphysiologic doses of microtubule poisons cause mitotic arrest and can activate the mitotic stopwatch. However, in physiologic concentrations of clinically useful microtubule poisons, cells proceed though mitosis and divide their chromosomes on mitotic spindles that are at least transiently multipolar. Though these low concentrations may result in a brief mitotic delay, it is substantially shorter than the arrest caused by high concentrations of microtubule poisons, and the one mimicked here by 16 hours of 0.4 mg/mL nocodazole or 48 hours of 100 nM paclitaxel. Resistance to mitotic arrest occurs through different mechanisms than resistance to multipolar spindles, raising concerns about the relevance of prolonged mitosis to paclitaxel response in cancer. Nocodazole is a microtubule poison that is not used clinically and does not induce multipolar spindles, so a similar apoptotic response to both drugs increases concern about a lack of physiological relevance. Moreover, clinical response to paclitaxel does not correlate with p53 status (PMID: 10951339, PMID: 8826941, PMID: 10955790). No evidence is presented that GMCL1 affects cellular response to clinically relevant doses of paclitaxel.

Comments on revisions:

(1) The claim that GMCL1 modulates paclitaxel sensitivity in cancer should be toned down. Inaccurate statements based on an outdated understanding of the anti-cancer mechanism of paclitaxel should be removed (eg lines 42-44: "In cancers that are resistant to paclitaxel, a microtubule-targeting agent, cells bypass mitotic surveillance activation, allowing unchecked proliferation...", lines 73-75: "Proper mitotic arrest is critical for the efficacy of microtubule-targeting therapies...", lines 78-79: "This resistance is frequently associated with loss of MSP activity, for example due to defective p53 signaling". As cited in the public review, p53 status does not correlate with paclitaxel response in cancer.)

(2) Perform timelapse experiments +/- GMCL1 siRNA in the absence of drug and in the presence of low, physiologically relevant concentrations of paclitaxel (5-10 nM), as well as supraphysiologic concentrations (100 nM) and correlate mitotic duration with cell cycle arrest. Test if co-depletion of 53BP1 with GMCL1 rescues cell cycle arrest after a substantially prolonged mitosis. Perform these experiments in a cell line with an intact mitotic stopwatch.

---

## [Author Response]

The following is the authors’ response to the original reviews.

**Public Reviews:**
**Reviewer #1(Public review)**:In this manuscript, Pagano and colleagues test the idea that the protein GMCL1 functions as a substrate receptor for a Cullin RING 3 E3 ubiquitin ligase (CUL3) complex. Using a pulldown approach, they identify GMCL1 binding proteins, including the DNA damage scaffolding protein 53BP1. They then focus on the idea that GMCL1 recruits 53BP1 for CUL3-dependent ubiquitination, triggering subsequent proteasomal degradation of ubiquitinated 53BP1.In addition to its DNA damage signalling function, in mitosis, 53BP1 is reported to form a stopwatch complex with the deubiquitinating enzyme USP28 and the transcription factor p53 (PMID: 38547292). These 53BP1-stopwatch complexes generated in mitosis are inherited by G1 daughter cells and help promote p53-dependent cell cycle arrest independent from DNA damage (PMID: 38547292). Several studies show that knockout of 53BP1 overcomes G1 cell cycle arrest after mitotic delays caused by anti-mitotic drugs or centrosome ablation (PMID: 27432897, 27432896). In this model, it is crucial that 53BP1 remains stable in mitosis and more stopwatch complex is formed after delayed mitosis.Major concerns:Pagano and coworkers suggest that 53BP1 levels can sometimes be suppressed in mitosis if the cells overexpress GMCL1. They carry out a bioinformatic analysis of available public data for p53 wild-type cancer cell lines resistant to the anti-mitotic drug paclitaxel and related compounds. Stratifying GMCL1 into low and high expression groups reveals a weak (p = 0.05 or ns) correlation with sensitivity to taxanes. It is unclear on what basis the authors claim paclitaxel-resistant and p53 wild-type cancer cell lines bypass the mitotic surveillance/timer pathway. They have not tested this. Figure 3 is a correlation assembled from public databases but has no experimental tests. Figure 4 looks at proliferation but not cell cycle progression or the length of mitosis. The main conclusions relating to cell cycle progression and specifically the link to mitotic delays are therefore not supported by experimental data. There is no imaging of the cell cycle or cell fate after mitotic delays, or analysis of where the cells arrest in the cell cycle. Most of the cell lines used have been reported to lack a functional mitotic surveillance pathway in the recent work by Meitinger. To support these conclusions, the stability of endogenous 53BP1 under different conditions in cells known to have a functional mitotic surveillance pathway needs to be examined. A key suggestion in the work is that the level of GMCL1 expression correlates with resistance to taxanes. For the mitotic surveillance pathway, the type of drug (nocodazole, taxol, etc) used to induce a delay isn't thought to be relevant, only the length of the delay. Do GMCL1-overexpressing cells show resistance to anti-mitotics in general?

We thank the reviewer for this insightful comment. We propose that GMCL1 promotes CUL3-dependent ubiquitination of 53BP1 during prolonged mitotic arrest, thereby facilitating its proteasome-dependent degradation. To evaluate the potential clinical relevance of this mechanism, we stratified cancer cell lines based on GMCL1 mRNA expression using publicly available datasets from DepMap (PMID: 39468210). We observed correlations between GMCL1 expression levels and taxane sensitivity that appear to reflect specific cancer type-drug combinations. To experimentally evaluate this correlation and obtain mechanistic insights, we performed knockdown experiments in hTERT-RPE1 cells, which are known to possess an intact mitotic surveillance pathway. Silencing of GMCL1 alone inhibited cell proliferation and induced apoptosis, while co-depletion of either TP53BP1 or USP28 significantly rescued these effects. These results suggest that GMCL1 modulates the stability of 53BP1 and therefore the availability of the 53BP1-USP28-p53 ternary complex in cells with a functional mitotic surveillance pathway (MSP) (new Figure 5I,J) directly linking GMCL1 to the regulation of the MSP complex. Moreover, to further support our mechanism, we assessed the effect of GMCL1 levels on cell cycle progression. Briefly, following nocodazole synchronization and release, we treated cells with EdU and performed FACS analyses at different times. Knockdown of GMCL1 alone led to a delayed cell cycle progression, but co-depletion of either TP53BP1 or USP28 restored this phenotype (new Figure 3A and new Supplementary Figure 3A-C). These results are consistent with our proliferation data and suggest that the observed effects of GMCL1 are specific to mitotic exit. Finally, overexpression of GMCL1 accelerates cell cycle progression (as assessed by FACS analyses) upon release from prolonged mitotic arrest (new Figure 3B and new Supplementary Figure 3D-E).

Importantly, if GMCL1 specifically degrades 53BP1 during prolonged mitotic arrests, the authors should show what happens during normal cell divisions without any delays or drug treatments. How much 53BP1 is destroyed in mitosis under those conditions? Does 53BP1 destruction depend on the length of mitosis, drug treatment, or does 53BP1 get degraded every mitosis regardless of length? Testing the contribution of key mitotic E3 ligase activities on mitotic 53BP1 stability, such as the anaphase-promoting complex/cyclosome (APC/C) is important in this regard. One previous study reported an analysis of putative APC/C KEN-box degron motifs in 53BP1 and concluded these play a role in 53BP1 stability in anaphase (PMID: 28228263).

Physiological mitosis under unperturbed conditions is typically brief (approximately 30 minutes), making protein quantification during this window challenging. Despite this, we tried by synchronizing cells using RO-3306 and releasing them into drug-free medium to assess GMCL1 dynamics during normal mitosis. Under these conditions, GMCL1 expression was similar to that in asynchronous cells and higher than the levels upon extended mitosis. However, when we attempted to measure the half-life of proteins using cycloheximide, most cells died, likely due to the toxic effect of cycloheximide in cells subjected to co-treatment with RO-3306 or nocodazole. This is the same reasons why in Figure 2C, we assessed 53BP1 in daughter cells rather than mitotic cells.

There is no direct test of the proposed mechanism, and it is therefore unclear if 53BP1 is ubiquitinated by a GMCL1-CUL3 ligase in cells, and how efficient this process would be at different cell cycle stages. A key issue is the lack of experimental data explaining why the proposed mechanism would be restricted to mitosis. Indirect effects, such as loss of 53BP1 from the chromatin fraction during M phase upon GMCL1 overexpression, do not necessarily mean that 53BP1 is degraded. PLK1-dependent chromatin-cytoplasmic shuttling of 53BP1 during mitotic delays has been described previously (PMID: 38547292, 37888778). These papers are cited in the text, but the main conclusions of those papers on 53BP1 incorporation into a stopwatch complex during mitotic delays have been ignored. Are the authors sure that 53BP1 is destroyed in mitosis and not simply re-localised between chromatin and non-chromatin fractions? At the very least, these reported findings should be discussed in the text.

To examine whether GMCL1 promotes 53BP1 ubiquitination in cells, we expressed in cells Trypsin-Resistant Tandem Ubiquitin-Binding Entity (TR-TUBE), a protein that binds polyubiquitin chains. Abundant, endogenous ubiquitinated 53BP1 co-precipitated with TR-TUBE constructs only when wild-type GMCL1 but not the E142K GMCL1 mutant, was expressed (new Figure 2D). The PLK1-dependent incorporation of 53BP1 into the stopwatch complex and the chromatin-cytoplasmic shuttling of 53BP1 during mitotic delays is now discussed in the text. That said, compared to parental cells, 53BP1 levels in the chromatin fraction are high in two different GMCL1 KO clones in M phase arrested cells (Figure 2A-B). This increase does not correspond to a decrease in the 53BP1 soluble fraction (Figure 2A and new Supplementary Figure 2D), suggesting decreased 53BP1 is not due to re-localization. The increased half-life of 53BP1 in daughter cells (Figure 2C), also supports this hypothesis.

The authors use a variety of cancer cell line models throughout their study, most of which have been reported to lack a functional mitotic surveillance pathway. U2OS and HCT116 cells do not respond normally to mitotic delays, despite being annotated as p53 WT. Other studies have used p53 wild-type hTERT RPE-1 cells to study the mitotic surveillance pathway. If the model is correct, then over-expressing GMCL1 in hTERT-RPE1 cells should suppress cell cycle arrest after mitotic delays, and GMCL1 KO should make the cells more sensitive to delays. These experiments are needed to provide an adequate test of the proposed model.

We greatly appreciate the reviewer’s suggestion regarding overexpression of GMCL1 in hTERT-RPE1 cells. To address this, we generated stable RPE1 cells expressing V5-tagged GMCL1 and conducted EdU incorporation assays following nocodazole synchronization and release. Overexpression of GMCL1 enhanced cell cycle progression compared to control cells (new Figure 3B and new Supplementary Figure 3D-E) after mitotic arrest, consistent with our model. We, therefore, propose that GMCL1 controls 53BP1 stability to suppress p53-dependent cell cycle arrest.

We also want to point out that while some papers suggest that HCT116 and U2OS cells do not have an intact mitotic surveillance pathway, others have shown that the MSP is indeed functioning in HCT116 cells and can be triggered with variable efficiency in U2OS cells (PMID: 38547292). This is likely due to high heterogeneity and extensive clonal diversity of cancer cell lines grown in different labs. Please see examples in PMIDs: 3620713, 30089904, and 30778230. In particular, PMID: 30089904 shows that this heterogeneity correlates with considerably different drug responses.

To conclude, while the authors propose a potentially interesting model on how GMCL1 overexpression could regulate 53BP1 stability to limit p53-dependent cell cycle arrest, it is unclear what triggers this pathway or when it is relevant. 53BP1 is known to function in DNA damage signalling, and GMCL1 might be relevant in that context. The manuscript contains the initial description of GMCL1-53BP1 interaction but lacks a proper analysis of the function of this interaction and is therefore a preliminary report.

We hope that the new experiments, along with the clarifications provided in this response letter and revised manuscript, offer the reviewer increased confidence in the robustness and validity of our proposed model.

**Reviewer #2 (Public review):**
This study investigates the role of GMCL1 in regulating the mitotic surveillance pathway (MSP), a protective mechanism that activates p53 following prolonged mitosis. The authors identify a physical interaction between 53BP1 and GMCL1, but not with GMCL2. They propose that the ubiquitin ligase complex CRL3-GMCL1 targets 53BP1 for degradation during mitosis, thereby preventing the formation of the "mitotic stopwatch" complex (53BP1-USP28-p53) and subsequent p53 activation. The authors show that high GMCL1 expression correlates with resistance to paclitaxel in cancer cell lines that express wild-type p53. Importantly, loss of GMCL1 restores paclitaxel sensitivity in these cells, but not in p53-deficient lines. They propose that GMCL1 overexpression enables cancer cells to bypass MSP-mediated p53 activation, promoting survival despite mitotic stress. Targeting GMCL1 may thus represent a therapeutic strategy to re-sensitize resistant tumors to taxane-based chemotherapy.Strengths:This manuscript presents potentially interesting observations. The major strength of this article is the identification of GMCL1 as a 53BP1 interaction partner. The authors identified relevant domains and showed that GMCL1 controls 53BP1 stability. The authors further show a potentially interesting link between GMCL1 status and sensitivity to Taxol.Weaknesses:However, the manuscript is significantly weakened by unsubstantiated mechanistic claims, overreliance on a non-functional model system (U2OS), and overinterpretation of correlative data. To support the conclusions of the manuscript, the authors must show that the GMCL1-dependent sensitivity to Taxol depends on the mitotic surveillance pathway.

To demonstrate that GMCL1-dependent taxane sensitivity is mediated through the mitotic surveillance pathway (MSP), we now performed experiments using hTERT-RPE1 (RPE1) cells, a widely used, non-transformed cell line known to possess a functional MSP. We compared RPE1 cells with knockdown of GMCL1 alone to those with simultaneous knockdown of GMCL1 and either TP53BP1 or USP28. Upon paclitaxel (Taxol) treatment, cells with GMCL1 knockdown exhibited suppressed proliferation and increased apoptosis. Notably, these phenotypes were rescued by co-depletion of TP53BP1 or USP28 (new Figure 5I,J). These results support the notion that GMCL1 contributes to MSP activity, at least in part, through its regulation of 53BP1.

To further strengthen our mechanistic experiments, we assessed the effect of GMCL1 levels on cell cycle progression. Following nocodazole synchronization and release, we treated cells with EdU and performed FACS analyses at different times. Knockdown of GMCL1 alone led to a delay in cell cycle progression, but co-depletion of either TP53BP1 or USP28 alleviate this phenotype (new Figure 3A and new Supplementary Figure 3A, B). These results are consistent with our proliferation data.

**Reviewer #3(Public review)**:Summary:In this study, Kito et al follow up on previous work that identified Drosophila GCL as a mitotic substrate recognition subunit of a CUL3-RING ubiquitin ligase (CRL3) complex.Here they characterize mutants of the human ortholog of GCL, GMCL1, that disrupt the interaction with CUL3 (GMCL1E142K) and that lack the substrate interaction domain (GMCL1 BBO). Immunoprecipitation followed by mass spectrometry identified 9 proteins that interacted with wild-type FLAG-GMCL1 and GMCL1 EK but not GMCL1 BBO. These proteins included 53BP1, which plays a well-characterized role in double-strand break repair but also functions in a USP28-p53-53BP1 "mitotic stopwatch" complex that arrests the cell cycle after a substantially prolonged mitosis. Consistent with the IP-MS results, FLAG-GMCL1 immunoprecipitated 53BP1. Depletion of GMCL1 during mitotic arrest increased protein levels of 53BP1, and this could be rescued by wild-type GMCL1 but not the E142K mutant or a R433A mutant that failed to immunoprecipitate 53BP1.Using a publicly available dataset, the authors identified a relatively small subset of cell lines with high levels of GMCL1 mRNA that were resistant to the taxanes paclitaxel, cabazitaxel, and docetaxel. This type of analysis is confounded by the fact that paclitaxel and other microtubule poisons accumulate to substantially different levels in various cell lines (DOI: 10.1073/pnas.90.20.9552 , DOI: 10.1091/mbc.10.4.947), so careful follow-up experiments are required to validate results. The correlation between increased GMCL1 mRNA and taxane resistance was not observed in lung cancer cell lines. The authors propose this was because nearly half of lung cancers harbor p53 mutations, and lung cancer cell lines with wild-type but not mutant p53 showed the correlation between increased GMCL1 mRNA and taxane resistance. However, the other cancer cell types in which they report increased GMCL1 expression correlates with taxane sensitivity also have high rates of p53 mutation. Furthermore, p53 status does not predict taxane response in patients (DOI: 10.1002/1097-0142(20000815)89:4<769::aid-cncr8>3.0.co;2-6 , DOI: 10.1002/(SICI)1097-0142(19960915)78:6<1203::AID-CNCR6>3.0.CO;2-A , PMID: 10955790).The authors then depleted GMCL1 and reported that it increased apoptosis in two cell lines with wild-type p53 (MCF7 and U2OS) due to activation of the mitotic stopwatch. This is surprising because the mitotic stopwatch paper they cite (DOI: 10.1126/science.add9528) reported that U2OS cells have an inactive stopwatch and that activation of the stopwatch results in cell cycle arrest rather than apoptosis in most cell types, including MCF7. Beyond this, it has recently been shown that the level of taxanes and other microtubule poisons achieved in patient tumors is too low to induce mitotic arrest (DOI: 10.1126/scitranslmed.3007965 , DOI: 10.1126/scitranslmed.abd4811 , DOI: 10.1371/journal.pbio.3002339), raising concerns about the relevance of prolonged mitosis to paclitaxel response in cancer. The findings here demonstrating that GMCL1 mediates degradation of 53BP1 during mitotic arrest are solid and of interest to cell biologists, but it is unclear that these findings are relevant to paclitaxel response in patients.Strengths:This study identified 53BP1 as a target of CRL3GMCL1-mediated degradation during mitotic arrest. AlphaFold3 predictions of the binding interface, followed by mutational analysis, identified mutants of each protein (GMCL1 R433A and 53BP1 IEDI1422-1425AAAA) that disrupted their interaction. Knock-in of a FLAG tag into the C-terminus of GMCL1 in HCT116 cells, followed by FLAG immunoprecipitation, confirmed that endogenous GMCL1 interacts with endogenous CUL3 and 53BP1 during mitotic arrest.Weaknesses:The clinical relevance of the study is overinterpreted. The authors have not taken relevant data about the clinical mechanism of taxanes into account. Supraphysiologic doses of microtubule poisons cause mitotic arrest and can activate the mitotic stopwatch. However, in physiologic concentrations of clinically useful microtubule poisons, cells proceed through mitosis and divide their chromosomes on mitotic spindles that are at least transiently multipolar. Though these low concentrations may result in a brief mitotic delay, it is substantially shorter than the arrest caused by high concentrations of microtubule poisons, and the one mimicked here by 16 hours of 0.4 mg/mL nocodazole, which is not used clinically and does not induce multipolar spindles. Resistance to mitotic arrest occurs through different mechanisms than resistance to multipolar spindles. No evidence is presented in the current version of the manuscript that GMCL1 affects cellular response to clinically relevant doses of paclitaxel.

We agree that it would be an overstatement to claim that GMCL1 and p53 regulates paclitaxel sensitivity in cancer patients in a clinical context. The correlations we observed were based on publicly available cancer cell lines from datasets catalogued in CCLE and DepMap, which do not fully account for clinical heterogeneity and patient-specific factors. In response to this important point, we have revised the text accordingly.

In the experiments shown in former Figure 4A-H (now Figure 5A-H) and in those shown in the new Figure 5I-J, we used 100 nM paclitaxel to test the hypothesis that low GMCL1 levels sensitizes cancer cells in a p53-dependent manner. Here, paclitaxel was chosen to mimic the conditions reported in the PRISM dataset (PMID: 32613204), which compiles the proliferation inhibitory activity of 4,518 compounds tested across 578 cancer cell lines. Consistent with our cell cycle findings, the paclitaxel sensitivity caused by GMCL1 depletion was reverted by silencing 53BP1 or USP28 (new Figure 5I-J), again supporting the involvement of the stopwatch complex. We are unsure about how to model the “physiologic concentrations of clinically useful microtubule poisons” in cell-based studies. A recent review notes that “The time above a threshold paclitaxel plasma concentration (0.05 mmol/L) is important for the efficacy and toxicity of the drug” (PMID: 28612269). Two other reviews mention that the clinically relevant concentration of paclitaxel is considered to be plasma levels between 0.05–0.1 μmol/L (approximately 50–100 nM) and that in clinical dosing, typical patient plasma concentrations after paclitaxel infusion range from 80–280 nM, with corresponding intratumoral concentrations between 1.1–9.0 μM, due to drug accumulation in tumor tissue (PMIDs: 24670687 and 29703818). We have now emphasized in the revised text the rationale for using 100 nM paclitaxel in our experiments.

**Recommendations for the authors:**
**Reviewer #1** (**Recommendations for the authors):**General comments on the Figures:(1) Western blots lack molecular weight markers on most panels and are often over-exposed and over-contrasted, rendering them hard to interpret.

We have now included molecular weight markers in all Western blot panels. We have also reprocessed the images to avoid overexposure and excessive contrast, ensuring that the bands are clearly visible and interpretable.

(2) Input and IP samples do not show percentage loading, so it is hard to interpret relative enrichments.

In the revised figures, we have indicated what % of the input was loaded.

(3) The authors change between cell line models for their experiments, and this is not clear in the figures. These are important details for interpreting the data, as many of the cell lines used are not functional for the mitotic surveillance pathway.

In the revised manuscript, we have clearly indicated the specific cell lines used in each experiment in the figure legends. Additionally, to address concerns regarding the mitotic surveillance pathway, we have included new experiments using hTERT-RPE1 cells, which have been reported to possess a functional mitotic surveillance pathway (MSP) (Figure 4I-J).

(4) No n-numbers are provided in the figure legends. Are the Western blots provided done once, or are they reproducible? Many of the blots would benefit from quantification and presentation via graphs to test for reproducible changes to 53BP1 levels under the different conditions.

As now indicated in the methods section, we have conducted each Western blot no less than three times, yielding results that exhibit a high degree of reproducibility. A representative Western blot has been selected for each figure. We did not include densiometric quantification of immunoblots, given that the semi-quantitative nature of this technique would lead to an overinterpretation of our data; unfortunately, this is a limitation of the technique. In fact, eLife and other similar scientific journals do not adhere to the practice of quantifying Western blots. One exception to this norm is for protein half-life studies, which is done to measure the kinetics of decay rates and their internal comparisons. Accordingly, the experiments in Figure 2C were quantified.

(5) Graphs displayed in the supplementary figures are blacked out, and individual data points cannot be visualised. All graphs should have individual data points clearly visible.

We revised the quantified graphs and replaced them with scatter plots to clearly display individual data points, showing sample distribution.

Additional experiments with specific comments on Figures:(1) Figure 1C-D: the relative amount of 53BP1 co-precipitating with FLAG-tagged GMCL1 WT appears very different between the two experiments. If the idea is that MLN4924 (Cullin neddylation inhibitor) makes the interaction easier to capture, then this should be explained in the text, and ideally shown on the same gel/blot -/+ MLN4924.

We now present the samples treated with and without MLN4924 on the same gel/blot to allow direct comparison (new Figure 1D) and clarified this point in the text.

(2) Figure 1E: The figure legend states that GMCL1 was immunoprecipitated, but the Figure looks as though FLAG-tagged 53BP1 was the bait protein being immunoprecipitated? Can the authors clarify?

We thank the reviewer for pointing out the discrepancy between the figure and the figure legend in Figure 1E. The immunoprecipitation was indeed performed using FLAG-tagged 53BP1, and we have now rectified the figure legend accordingly.

(3) Figure 1F: Rather than parental cell lysate, the better control would be to IP FLAG from another FLAG-tagged expressing cell line, to rule out non-specific binding with the FLAG tag at the non-overexpressed level.

Figure 1F shows interaction at the endogenous level. The specificity of binding with overexpressed proteins is shown in Figures 1C and 1D.

The USP28 blot is over-exposed and makes it hard to see any changes in electrophoretic mobility - it looks as though there is a change between the parental and the KI cell line? It is surprising that USP28 would co-IP with GMCL1 (presumably because USP28 is bound to 53BP1) if the function of GMCL1-53BP1 interaction is to promote 53BP1 degradation. Can the authors reconcile this? Crucially, if the authors claim that the 53BP1-GMCL1 interaction is specific to prolonged mitosis, then this experiment should be repeated and performed with asynchronous, normal-length mitosis, and prolonged mitosis conditions. This is vital for supporting the claim that this interaction only occurs during prolonged mitoses and does not occur in every mitosis regardless of length.

This is a good point. Unfortunately, many of the protein-protein interactions occur post lysis. Therefore, we could not observe differences in asynchronous vs. mitotic cells.

(4) Figure S1F: Label on blot should be CUL3 not CUI3.

We thank the reviewer for pointing this out and we have corrected the typo.

(5) Figure 2A: The authors suggest an increase in chromatin-bound 53BP1 in GMCL1 KO U2OS cells, specifically in M phase. Again, is this time in mitosis dependent, or would this be evident in every mitosis, regardless of length? Such an experiment would benefit from repetition and quantification to test whether the observed effect is reproducibly consistent. If the authors' model is correct, simply treating U2OS WT mitotic cells with MG132 during the mitotic arrest and performing the same fractionation should bring 53BP1 levels up to that seen in GMCL1 KO cells under the same conditions.

The reviewer’s suggestion to assess 53BP1 accumulation in wild-type U2OS cells treated with MG132 during mitotic arrest is indeed highly relevant. However, treatment with MG132 during prolonged mitosis consistently led to significant cell death, making it technically challenging to evaluate 53BP1 levels under these conditions.

(6) Figure 2B: The authors restore GMCL1 expression in the KO U2OS cells using WT and 2 distinct mutant cDNAs. However, the expression of these constructs is not equivalent, and thus their effects cannot be directly compared. It is also surprising that GMCL1 is much higher in M phase samples in this experiment (shouldn't it be destroyed?), when no such behaviour has been observed in the other figures.

There is no evidence in our study or others that GMCL1 should be destroyed in M phase. We show that the R433A mutant is expressed at a level very similar to the WT protein, yet it doesn’t promote the degradation of 53BP1. It is true that the E142K is expressed less in mitotic cells whereas is the most expressed in asynchronous cells. For some reason, this mutant has an inverse behavior compared to the WT, limiting the interpretation of this result. We now mention this in the text.

(7) Figure 2C: The CHX experiment would benefit from inclusion of a control protein known to have a short half-life (e.g. c-myc, p53). Is GMCL1 known to have a relatively short half-life? It looks as though GMCL1 disappears after 1 h CHX treatment (although hard to definitively tell in the absence of molecular weight markers). 53BP1 appears to continue declining in the absence of GMCL1, which is surprising if p53BP1 degradation requires GMCL1. How can the authors reconcile this?

As a control for the CHX chase experiments, we included p21, whose protein levels decreased in a CHX-dependent. GMCL1 itself also appeared to undergo degradation upon CHX treatment, but it doesn’t disappear completely.

(8) Supplemental Figure 2:Transcription is largely inhibited in M phase, so the p53 target gene transcripts present in M phase are inherited from the preceding G2 phase. The qPCR's thus need a reference sample to compare against. I.e., was p21/PUMA/NOXA mRNA already low in G2 in the GMCL1 KO + WT cells before they entered mitosis? Or is the mRNA stability affected during M phase specifically? Is this effect on the mRNA dependent on the time in mitosis?

It is well established that transcription is not entirely shut down during mitosis, particularly for a subset of genes involved in cell cycle regulation. For example, p21, PUMA, NOXA, and p53 mRNAs have been shown to remain actively transcribed during mitosis (see Table S5 in PMID: 28912132). However, we currently lack direct evidence that p53 activation during mitosis, specifically through the mitotic surveillance pathway, drives the transcription of p21, PUMA, or NOXA mRNAs during M phase. In the absence of such mechanistic data, we opted to exclude these analyses from the final figures.

Panel B: blots are too over-exposed to see differences in p53 stability under the different conditions. Mitotic samples should be included to show how these differ from the G1 samples.

The background of all blot images has been adjusted to ensure clarity and consistency.

Panel D: The authors show no significant difference in the cell cycle profiles of the GMCL1 KO and reconstituted cells compared to parental U2OS cells. This should also be performed in the G1 daughter cells following a prolonged mitosis, to test the effect of the different GMCL1 constructs on G1 cell cycle arrest. U2OS cells have been reported not to have a functional mitotic surveillance pathway (Meitinger et al, Science, 2024), so U2OS cells are perhaps not a good model for testing this.

We performed cell cycle profiling using EdU incorporation in hTERT-RPE1 cells, which possess a functional MSP, to evaluate cell cycle progression in daughter cells following prolonged mitosis. We observed that GMCL1 knockdown alone leads to G1-phase arrest. In contrast, co-depletion of GMCL1 with either 53BP1 or USP28 bypasses this arrest, indicating that GMCL1 regulates cell cycle progression in an MSP-dependent manner. Please see also the answer to the public review above.

(9) Figure 3:The authors show expression data for GMCL1 in the different cancer cell lines. This should be validated for a subset of cancer cell lines at the GMCL1 protein level, and cross-correlated to their MSP/mitotic timer status. Does GMCL1 depletion or knockout in p53 wild-type cancer cell lines overexpressing GMCL1 protein restore mitotic surveillance function?

We were unable to assess GMCL1 protein levels using publicly available proteomics datasets, as GMCL1 expression was not detected. In p53 wild-type hTERT-RPE1 cells, GMCL1 knockdown impaired the mitotic surveillance pathway, as evidenced by G1-phase arrest following prolonged mitosis (new Figure 3A and new Supplementary Figure 3A, B). This arrest was rescued by co-depletion of either TP53BP1 or USP28, indicating that GMCL1 acts upstream of the MSP.

(10) Figure 4:The authors show siRNA experiments depleting GMCL1 and testing the effects of GMCL1 loss on cell viability and apoptosis induction. This is performed in different cell line backgrounds. However, there is no demonstration that any of the observed effects are due to a lack of GMCL1 activity on 53BP1. These experiments need to be repeated in 53BP1 co-depleted cells to test for rescue. Without this, the interpretation is purely correlative.

We assessed the effects of GMCL1 knockdown, alone or in combination with TP53BP1 or USP28 knockdown, on cell viability and apoptosis in hTERT-RPE1 cells using siRNA. Knockdown of GMCL1 alone led to a significant reduction in cell viability and an increase in apoptosis. However, co-depletion of GMCL1 with either TP53BP1 or USP28 restored both cell viability and apoptosis levels to those observed in control cells (new Figure 5I,J).

(11) Text comments:Line 257: HeLa cells supress p53 through the E6 viral protein and are not "mutant" for p53.The authors should cite early work by Uetake and Sluder describing the effects of spindle poisons on the mitotic surveillance pathway.

We appreciate the reviewer’s comments – We have now made the necessary corrections.

**Reviewer #2 (Recommendations for the authors**):Major Points:(1) Unsubstantiated Mechanistic Claims:In Figures 3 and 4, the authors show correlations between GMCL1 expression and sensitivity to Taxol. However, they fail to demonstrate that the mitotic stopwatch is mechanistically involved. To support this conclusion, the authors must test whether deletion of 53BP1, USP28, or disruption of their interaction rescues Taxol sensitivity in GMCL1-depleted cells. Since 53BP1 also plays a role in DNA damage response, such rescue experiments are necessary to distinguish between mitotic surveillance-specific and broader stress-response effects. Deletion of USP28 would be particularly informative.

We sought to experimentally determine whether GMCL1 is involved in regulating the mitotic stopwatch. Knockdown of GMCL1 alone resulted in reduced cell proliferation and increased apoptosis. In contrast, co-depletion of GMCL1 with either TP53BP1 or USP28 restored both proliferation and apoptosis levels to those observed in control cells (new Figure 5I, J). To further strengthen our mechanistic experiments, we assessed the effect of GMCL1 levels on cell cycle progression. We conducted EdU incorporation assays following nocodazole synchronization and release. Knockdown of GMCL1 alone led to a delay in G1 progression, whereas co-depletion of either TP53BP1 or USP28 rescued normal cell cycle progression (new Figure 3A and new Supplementary Figure 3A, B). These results are consistent with our proliferation data and suggest that GMCL1 functions upstream of the ternary complex, likely by regulating 53BP1 protein levels.

(2) Model System Limitations (U2OS Cells):The use of U2OS cells is highly problematic for investigating the mitotic surveillance pathway. U2OS cells lack a functional mitotic stopwatch and do not arrest following prolonged mitosis in a 53BP1/USP28-dependent manner (PMID: 38547292). Therefore, conclusions drawn from this model system about the function of the mitotic surveillance pathway are not substantiated. Key experiments should be repeated in a cell line with an intact pathway, such as RPE1.

We now performed all key experiments also hTERT-RPE1 cells (see above). We also would like to point out that while some papers suggest that HCT116 and U2OS cells do not have an intact mitotic surveillance pathway, others have showed that the MSP is indeed functioning in HCT116 cells and can be triggered with variable efficiency in U2OS cells (PMID: 38547292). This is likely due to high heterogeneity and extensive clonal diversity of cancer cell lines grown in different labs. Please see examples in PMIDs: 3620713, 30089904, and 30778230. In particular, PMID: 30089904 shows that this heterogeneity correlates with considerably different drug responses.

(3) Misinterpretation of p53 Activity Timing:The manuscript states that "GMCL1 KO cells led to decreased mRNA levels of p21 and NOXA during mitosis" (line 194). However, it is well established that the mitotic surveillance pathway activates p53 in the G1 phase following prolonged mitosis-not during mitosis itself (PMID: 38547292). Therefore, the observed changes in mRNA levels during mitosis are unlikely to be relevant to this pathway.

We currently lack direct evidence that p53 activated during mitosis through the mitotic surveillance pathway directly influences the transcription of p21, PUMA, or NOXA mRNAs during M phase. Therefore, we have chosen to exclude these data from the final figures.

(4) Incorrect Interpretation of 53BP1 Chromatin Binding:The authors claim that 53BP1 remains associated with chromatin during mitosis, which contradicts established literature. It is known that 53BP1 is released from chromatin during mitosis via mitosis-specific phosphorylation (PMID: 24703952), and this is supported by more recent findings (PMID: 38547292). A likely explanation for the discrepancy may be contamination of mitotic fractions with interphase cells. The chromatin fraction data in Figure 2C must be interpreted with caution.

Our method to synchronize in M phase is rather stringent (see Supplementary Figure 3D as an example). The literature indicates that the bulk of 53BP1 is released from chromatin during mitosis. Yet, even in the two publications mentioned by the reviewer, there is a difference in the observable amount of 53BP1 bound to chromatin (compare Figure 2B in PMID: 38547292 and Figure 5A in PMID: 24703952). The difference is likely due to the different biochemical approaches used to purify chromatin bound proteins (salt and detergent concentrations, sonication, etc.). Using our fractionation approach, we can reliably separate the soluble fraction (containing also the nucleoplasmic fraction) and chromatin associated proteins as indicated by the controls such as a-Tubulin and Histon H3. We have now mentioned these limitations when comparing different fractionation methods in our discussion section.

(5) Inadequate Citation of Foundational Literature:The literature on the mitotic surveillance pathway is relatively limited, and it is essential that the authors provide a comprehensive and accurate account of its development. The foundational work by the Sluder lab (PMID: 20832310), demonstrating a p53-dependent arrest following prolonged mitosis, must be cited. Furthermore, the three key 2016 papers (PMID: 27432896, 27432897, 27432896) that identified the involvement of USP28 and 53BP1 in this pathway are critical and should be cited as the basis of the mitotic surveillance pathway.In contrast, the manuscript currently emphasizes publications that either contribute minimally or have been contradicted by prior and subsequent work. For example: PMID: 31699974, which proposes Ser15 phosphorylation of p53 as critical, has been contradicted by multiple groups (e.g., Holland, Oegema, and Tsou labs).PMID: 37888778, which suggests that 53BP1 must be released from kinetochores, is inconsistent with findings that indicate kinetochore localization is not relevant.The authors should thoroughly revise the Introduction to reflect what this reviewer would describe as a more accurate and scholarly approach to the literature.

We have substantially revised both the Introduction and Discussion sections to incorporate important references kindly suggested by the reviewer.

Minor Points:(1) Overexposed Western Blots:The Western blots throughout the manuscript are heavily overexposed and saturated, obscuring differences in protein levels and hindering data interpretation. The authors should provide properly exposed blots with quantification where appropriate.

We have provided Western blot images with appropriate exposure levels and included quantification where appropriate (i.e., to measure the kinetics of decay rates as in Figure 2C). For all the other immunoblots, we did not include densiometric quantification, given that the semi-quantitative nature of this technique would lead to overinterpretation of our data. This is, unfortunately, a limitation of the technique. In fact, eLife and other similar scientific journals do not adhere to the practice of quantifying Western blot analyses.

(2) Missing information in the graphs in Figure 2C and 4; S2? How many repeats? What are the asterisks?

Panels referenced above have been repeated several times, and further details are now provided in the figure legends.

**Reviewer #3 (Recommendations for the authors):**
(1) The claim that GMCL1 modulates paclitaxel sensitivity in cancer should be toned down

.

We agree that it would be an overstatement to claim that GMCL1 regulates paclitaxel sensitivity in cancer patients in a clinical context. The correlations we observed were based on publicly available, cell line–based datasets, which do not fully account for clinical heterogeneity and patient-specific factors. In response to this important point, we have revised our statements and corresponding text accordingly. We now placed greater emphasis on our molecular and cell biology studies.

(2) Additional experiments in low, physiologically relevant concentrations of paclitaxel would be interesting. It is possible that these concentrations activate the mitotic stopwatch in a portion of cells, in addition to inducing cell death due to chromosome loss, activation of an immune response, and chromothripsis. Results should be interpreted in the context of this complexity.

Please see the response to the public review.

(3) It would be helpful to show that CUL3 interacts with 53BP1 only in the presence of GMCL1.

We show that the binding of 53BP1 to GMCL1 is independent of the ability of GMCL1 to bind CUL3 (Figure 1C, D). The binding between 53BP1 and CUL3 is difficult to detect (Figure 1F) likely because it’s not direct but mediated by GMCL1.

(4) The GMCL1 "KO" lines appear to still express a low level of GMCL1 (Figure 2A), which should be acknowledged

We have included the GMCL1 mRNA expression data, as measured by RT-PCR, in Supplementary Figure 1G, demonstrating that GMCL1 expression was undetectable under the tested conditions.

(5) Additional description of the methods is warranted. This is particularly true for the database analysis that forms the basis for the claim that GMCL1 overexpression causes resistance to paclitaxel and other taxanes presented in Figure 3, the methodology used to obtain M-phase cells, and the concentration and duration of taxol treatment.

We have now extensively revised the Methods section.

(6) "Taxol" and "paclitaxel" are used interchangeably throughout the manuscript. Consistency would be preferable.

We have revised the manuscript to maintain consistency in the use of the terms “Taxol” and “paclitaxel” and now refer to “paclitaxel” when discussing that individual compound; “taxanes” when referring collectively to cabazitaxel, docetaxel and paclitaxel; and “Taxol” has been removed entirely to avoid redundancy or confusion.

(7) It is unclear why it is claimed that GMCL1 interacts "specifically" with 53BP1 (line 176) since multiple interactors were identified in the IP-MS study

We meant that the GMCL1 R433A mutant loses its ability to bind 53BP1, suggesting that the GMCL1-53BP1 interaction is not an artifact. We have now clarified the text.

(8) The bottom row in Figure S3 is misleading. Paclitaxel is not uniformly effective in every tumor of any given type, and so resistance occurs in every cancer type.

We fully agree that cancer is highly heterogeneous and that paclitaxel efficacy varies across tumors, even within the same histological subtype. Our intension was not to suggest uniform sensitivity/resistance, but rather to provide a high-level overview using aggregated data. We acknowledge that this coarse-grained representation may unintentionally imply overly generalized conclusions. To avoid potential misinterpretation, we have removed the corresponding panel in the revised paper.